# Design of target specific peptide inhibitors using generative deep learning and molecular dynamics simulations

Sijie Chen[1,8], Tong Lin[2,3,8], Ruchira Basu[4], Jeremy Ritchey[4], Shen Wang [1], Yichuan Luo[5], Xingcan Li[6], Dehua Pei [4] ✉, Levent Burak Kara[2] ✉ & Xiaolin Cheng [1,7] ✉

We introduce a computational approach for the design of target-specific peptides. Our method integrates a Gated Recurrent Unit-based Variational Autoencoder with Rosetta FlexPepDock for peptide sequence generation and binding affinity assessment. Subsequently, molecular dynamics simulations are employed to narrow down the selection of peptides for experimental assays. We apply this computational strategy to design peptide inhibitors that specifically target $\beta$-catenin and NF-$\kappa$B essential modulator. Among the twelve $\beta$-catenin inhibitors, six exhibit improved binding affinity compared to the parent peptide. Notably, the best C-terminal peptide binds $\beta$-catenin with an $IC_{50}$ of $0.010 \pm 0.06$ μM, which is 15-fold better than the parent peptide. For NF-$\kappa$B essential modulator, two of the four tested peptides display substantially enhanced binding compared to the parent peptide. Collectively, this study underscores the successful integration of deep learning and structure-based modeling and simulation for target specific peptide design.

The computational design of peptides for exploration of vast amino acid sequence spaces has gained popularity in antibiotics discovery and biomaterial design. Despite progress, designing peptide inhibitors that target-specific protein-protein interactions (PPIs) remains a significant challenge. Two broad categories of computational approaches, namely structure-based and sequence-based methods, have emerged for this purpose[1–4]. Structure-based approaches start design from a protein pocket or an existing peptide motif bound to the protein. For instance, Rosetta FlexPepDock, employing extensive conformational search and a template-based strategy, has demonstrated effectiveness in modeling diverse peptide–protein complexes[5–7]. Rooklin et al. proposed a method to identify pockets near the peptide motif and design inhibitors that optimize pocket occupation[8]. However, starting from a structural template may introduce biases or constraints in the sequence search, potentially resulting in suboptimal solutions[9].

Moreover, the computational cost associated with structure-based peptide design, involving tasks such as peptide structure prediction, docking and binding energy evaluation, poses scalability challenges for extensive peptide libraries. Particularly, the dynamic nature of peptide–protein interactions and the conformational flexibility of proteins as well as the need for considering solvent effects make the accurate prediction of binding poses a daunting task. Molecular Dynamics (MD) simulations have been increasingly utilized in peptide design[10]. While Rosetta FlexPepDock[11] samples efficiently the space of possible peptide conformations and rigid-body orientations on a given target protein surface, MD simulations offer a dynamic and detailed view of peptide–protein interactions at the atomic level,

[1]College of Pharmacy, The Ohio State University, 281 W Lane Ave, Columbus, OH, USA. [2]Mechanical Engineering Department, Carnegie Mellon University, 5000 Forbes Ave, Pittsburgh, PA, USA. [3]Machine Learning Department, Carnegie Mellon University, 5000 Forbes Ave, Pittsburgh, PA, USA. [4]Department of Chemistry and Biochemistry, The Ohio State University, 281 W Lane Ave, Columbus, OH, USA. [5]Electrical and Computer Engineering Department, Carnegie Mellon University, 5000 Forbes Ave, Pittsburgh, PA, USA. [6]Department of Radiology, Affiliated Hospital and Medical School of Nantong University, 20 West Temple Road, Nantong, Jiangsu, China. [7]Translational Data Analytics Institute, The Ohio State University, 1760 Neil Ave, Columbus, OH, USA. [8]These authors contributed equally: Sijie Chen, Tong Lin. ✉e-mail: pei.3@osu.edu; lkara@cmu.edu; cheng.1302@osu.edu

providing a rigorous framework for binding pose refinement and affinity calculation[12–15].

On the other hand, Recurrent Neural Network (RNN)-based Variational Autoencoder (VAE) models that offer a robust framework for sequence analysis and optimization have emerged as a cutting-edge approach in protein and peptide design. VAE, known for automated text processing and generation[16], has found success in modeling latent spaces of sequential data, particularly in language translation[17–19]. In peptide design, VAEs are trained to represent amino acid sequences in a continuous latent space, which, combined with a sampling method, such as the Metropolis Hasting (MH) algorithm, allow the generation of peptides with desired properties. Lim et al. utilized a conditional VAE to generate simplified molecular-input line-entry system (SMILES) strings of molecules with targeted chemical properties[20]. Das et al. combined VAE and MD simulations to generate experimentally validated antimicrobial peptides, demonstrating significant time savings in the drug discovery process[21]. Transformer models have also emerged as powerful tools in protein and peptide design, leveraging attention mechanisms to capture long-range dependencies and contextual information within sequences[22]. Notably, AlphaFold has demonstrated success in identifying high-affinity peptide binders from a given set of peptides[23–25]. However, its computational demands and limitations to natural amino acids poses challenges for sequence searches and compatibility with popular peptide affinity maturation strategies.

Compared to structure-based methods, sequence-based approaches are more likely to face constraints stemming from the limited availability of peptide–protein-binding data, leading to a more limited adoption in peptide design. As of 2021, the PDB database contains 186,892 peptide-containing protein structures, of which only 13,299 entries provide detailed information on peptide–protein interactions[26]. To overcome this limitation, a protein generative pre-trained transformer (GPT) model was devised to generate authentic protein sequences based on a small labeled dataset[27]. This strategy resembles that in natural language processing (NLP), where a generative model is pre-trained with unlabeled data and fine-tuned for a specific task[28].

In this study, we present a multi-step sequence generation algorithm that combines a deep learning-based generative model with structure-based modeling and simulation to efficiently generate high-affinity peptide binders targeting specific protein surfaces. The first step of our model involves a Gated Recurrent Unit (GRU)-based VAE and the Metropolis Hasting (MH) sampling algorithm to generate potential peptide sequences. Termed the VAE-MH process, this step effectively reduces the sequence search space from millions or billions to hundreds. The second step involves the rapid binding assessment of the VAE-MH-generated peptides using physics-based methods to fine-tune these peptides towards binding specific proteins. To balance speed and accuracy, peptide binding affinity is evaluated in a hierarchical manner: rank-ordering peptides using Rosetta FlexPepDock, followed by re-evaluation of high-ranked peptides with MD simulation. Specifically, each peptide generated by the VAE-MH process is superimposed onto a template structure bound to the target protein. The resulting peptide–protein complex structure is refined, allowing full flexibility to the peptide backbone and side chains, and its binding score is evaluated using Rosetta peptide–protein scoring functions. Subsequently, each high-ranked complex obtained from FlexPepDock undergoes binding energy calculation with the molecular mechanics/generalized Born surface area (MM/GBSA) method[29]. We demonstrate the effectiveness of our computational models by designing peptide inhibitors targeting $\beta$-catenin and Nuclear Factor (NF)-$\kappa$B essential modulator (NEMO), yielding promising results in testing with fluorescence-based binding assays.

## Results

### Improving $\beta$-catenin binding by peptide extension

The canonical Wnt/$\beta$-catenin signaling pathway regulates cell proliferation primarily through $\beta$-catenin (Fig. 1a)[30]. Therefore, disrupting the interaction between $\beta$-catenin and Wnt effectors, such as T-cell factor/lymphoid enhancer factor (TCF/LEF) represents a promising strategy to curb $\beta$-catenin hyperactivity and inhibit cell proliferation[31]. Various approaches, including hydrocarbon- and thioether-stapled peptides, have been developed to specifically target $\beta$-catenin and interfere with the $\beta$-catenin/TCF interaction[32]. For instance, the hydrocarbon-stapled peptide StAX-35, mimicking Axin, exhibited binding to $\beta$-catenin ($K_d = 0.013 \pm 0.002 \, \mu M$) and inhibited cell proliferation at 10s $\mu M$ concentrations. Using Axin as the template, Diderich et al. employed phage display to select thioether-stapled peptides with $K_d$ as low as 5.2 nM[33]. However, these peptides exhibited limited biological activity due to low cell permeability and short half-life time. Additionally, researchers have explored bicyclic $\beta$-sheet, $\beta$-hairpin and macrocyclic peptidomimetics to target $\beta$-catenin[34,35]. Schneider et al. employed the Rosetta suite of protein design tools to identify peptoid–peptide macrocycles capable of binding $\beta$-catenin and inhibiting the $\beta$-catenin–TCF interaction[36]. Although these peptidomimetics showed improved binding affinity, their biological activity remained modest, likely due to poor cell permeability. Given the availability of numerous peptide inhibitors, $\beta$-catenin could serve as an ideal system to validate our generative model. Our objective was to design N- and C-terminal extensions for a previously reported peptidyl inhibitor, Peptide 9 (Supplementary Table 1), with the aim of enhancing potency against the $\beta$-catenin/TCF interaction[37].

Superimposition of Peptide 9 onto the structure of a previously reported stapled $\alpha$-helical peptide StAX-35R bound to $\beta$-catenin (PDBid: 4DJS[32]) reveals that the $\beta$-catenin-binding cleft beneath Peptide 9 has a length of 24–28 Å, which is 8–12 Å longer than the peptide (Fig. 1b, c). Particularly, a void is observed at the N-terminus of $\beta$-catenin, allowing accommodation of an extra peptide fragment. These observations suggest that extending the $\alpha$-helical peptide by 4–7 residues on either N- or C-terminus could potentially enhance the interaction of the peptide with $\beta$-catenin. Peptide 9 was derived from a highly potent but membrane-impermeable peptidyl inhibitor of $\beta$-catenin, FAM-GGYPECILDCHLQRVIL-NH$_2$ ($K_d = 0.018 \, \mu M$)[33]. It was shown that the modified Peptide 9 was rendered highly cell-permeable, but bound to $\beta$-catenin with reduced affinity (IC$_{50}$ = 0.15 $\pm$ 0.04 $\mu M$)[37]. The conjugation to a cyclic cell-penetrating peptide was shown to have a minimal effect on the peptide-$\beta$-catenin binding, so the decreased binding seems to arise primarily from the replacement of the two cysteine residues with an aspartic acid and a lysine that were stapled with a DK linker as well as the removal of the N-terminal FAM dye. To this end, we aimed to improve the binding affinity of Peptide 9 for $\beta$-catenin by generating N- or C-terminal extensions composed solely of natural amino acids.

Further inspection of the Peptide 9-$\beta$-catenin-binding mode indicates that the two N-terminal glycine residues do not interact with the protein, but could increase the flexibility of the peptide, impairing its binding potency and specificity. Without the two glycine residues, a similar Peptide 14 shows slightly higher affinity (IC$_{50}$ = 0.11 $\mu M$) (Supplementary Table 1). Therefore, we removed the two glycines from Peptide 9 and chose the resulting peptide YPEDILDKHLQRVIL as the base model for extension (also referred to as the parent peptide below).

To test the idea of binding affinity maturation by terminal extension, we first employed Rosetta Design[38,39] to add 2–7 residues at either N- or C-terminus of the parent peptide and used Rosetta FlexPepDock[11] to evaluate the binding energies of the extended peptides using three metrics, including interface energy (I_sc), root-mean-square of interface atoms (rmsAll_if), and buried surface area of the interface (I_bsa). We report the differences in the three metrics

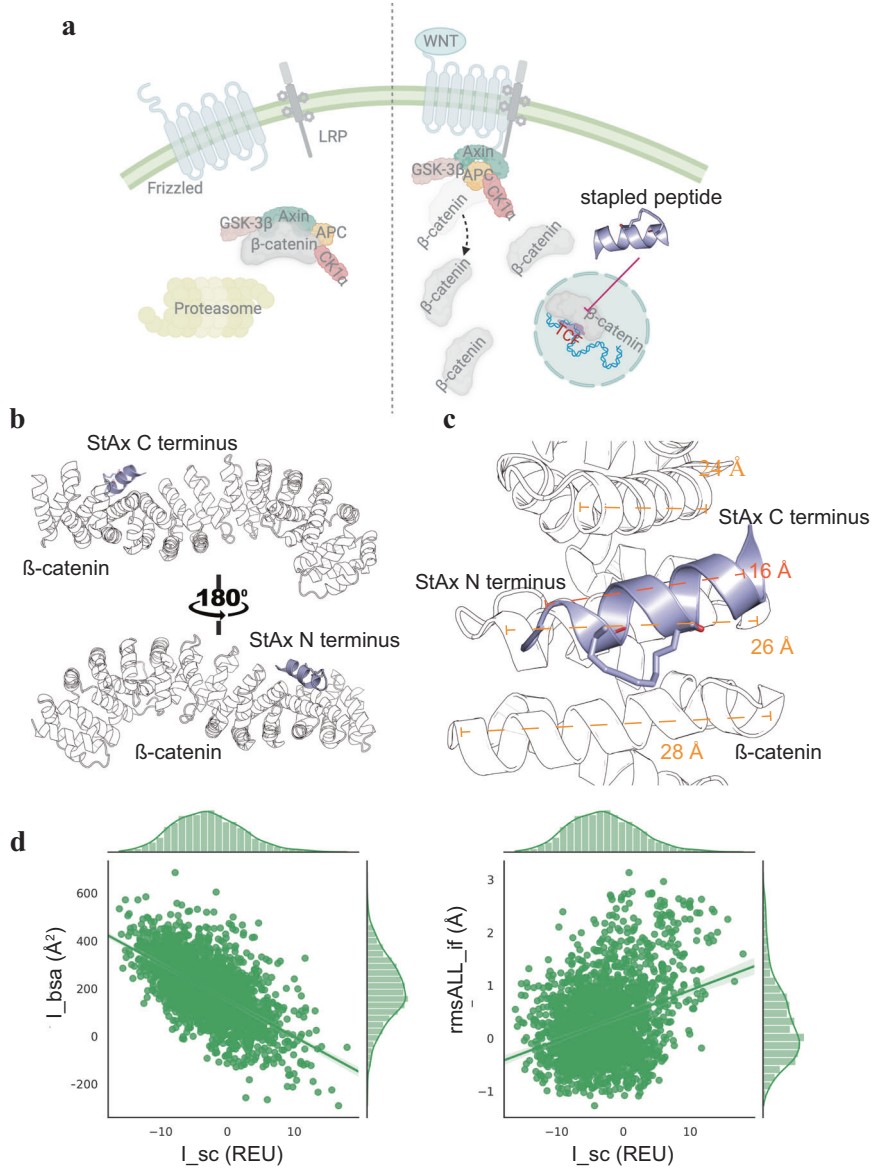

**Fig. 1 | Illustrations of the Wnt/β-catenin pathway and the binding mode of β-catenin and a stapled peptide. a** Wnt signal on/off pathways. Cartoons were created with BioRender.com. **b** Crystal structure of a stapled peptide StAX-35R bound to β-catenin viewed from the N- or C-terminus of the peptide (PDBid: 4DJS[32]). **c** Helical length comparisons of the stapled peptide with the three interacting helices of β-catenin. **d** Correlation plots of I_bsa vs. I_sc and rmsALL_if vs. I_sc for 20 Rosetta Design generated N- or C-terminal extensions. For each peptide, 100 conformations were sampled, with each conformation represented by a scatter on the scatterplot. The sample sizes in (**d**) are n = 2000. Source data are provided as a Source Data file.

between the extensions and the parent peptide in Supplementary Fig. 1. Overall, a large fraction of the extended peptides showed more favorable binding than the parent peptide, as evidenced by the presence of many negative values for I_sc and rmsAll_if and many positive values for I_bsa. In addition, as shown in Fig. 1d, the binding scores (I_sc) appear to correlate well with the interface contact areas (I_bsa). These results suggest that the improved binding affinity comes from the additional contacts formed between the extended peptides and the protein, substantiating the feasibility of terminal extension for affinity maturation of Peptide 9.

## VAE-MH peptide extension
Our goal is to extend the parent peptide at the N- or C-terminus by up to seven amino acids to obtain peptides with improved binding affinity for β-catenin. We call these added amino acids peptide extensions, which have over billions of combinatorial possibilities even if only the 20 natural amino acids are considered. To reduce this enormous

search space, we designed a latent space sampling algorithm comprising two components to generate peptide extensions that are more likely to be strong PPI binders (Fig. 2a).

The first component is to use a variational autoencoder (VAE) to represent peptides in a latent space. We prepared an unlabeled dataset for VAE training. This dataset combines protein sequences from three public databases, Uniprot[40], PixelDB[41], and THPdb[42]. Sequences longer than 22 residues were removed, yielding a dataset of around 4 million peptide sequences. We used VAE to embed a peptide sequence into an encoding, a concise continuous latent space vector that can be decoded back to the original representation of the sequence. The structure of the VAE is shown in Fig. 2b. The encoder module consists of an embedding network, a gated recurrent unit (GRU) and linear layers to output the mean and variance of the learned encodings. The Kullback−Leibler (KL) divergence is used as a distribution discrepancy measure between the encoding and a Gaussian distribution, which serves as a regularization term for the encoding distribution. The mean

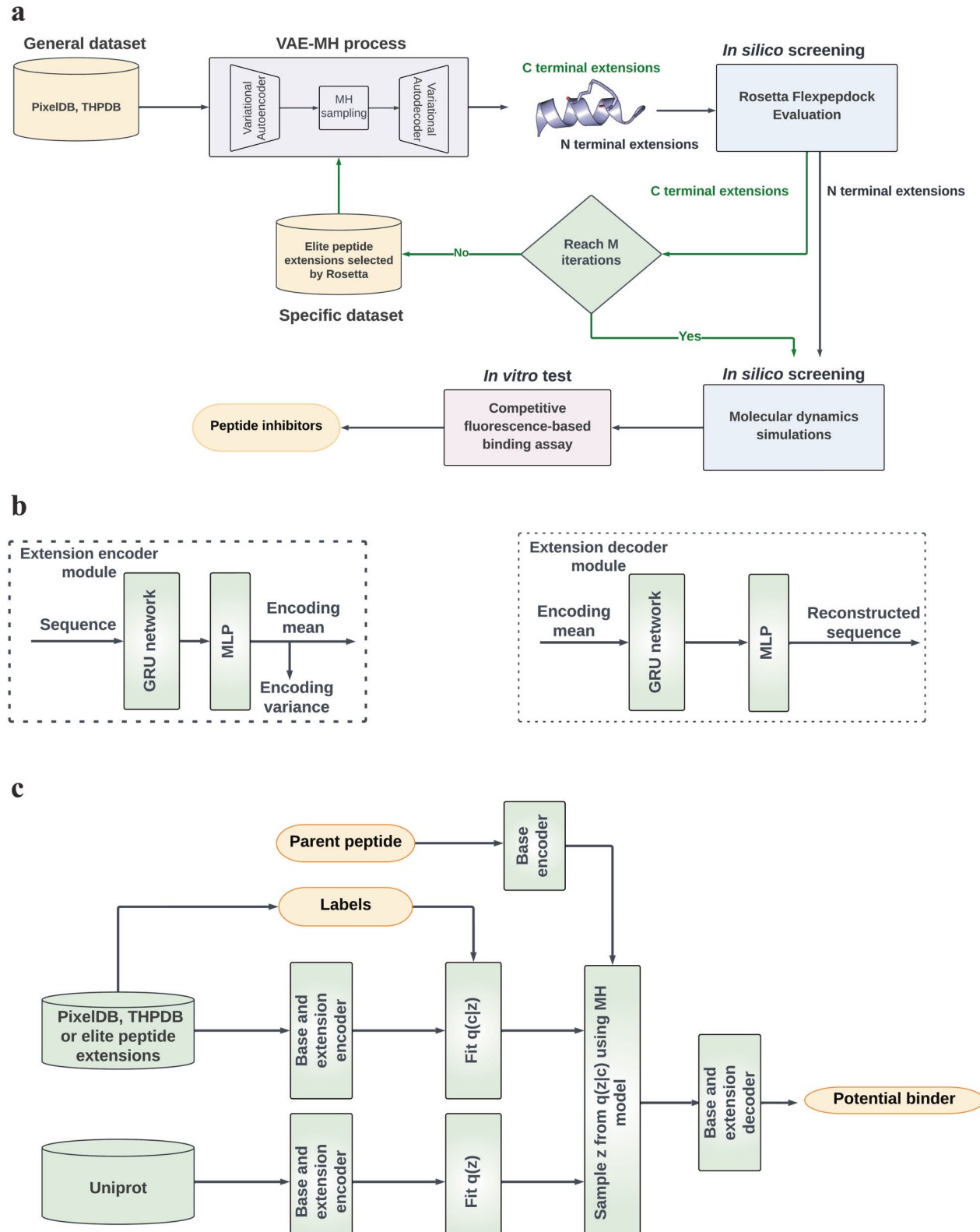

**Fig. 2 | Overview of the peptide design methodology. a** Workflow illustrating the design of β-catenin specific peptide inhibitors. The black arrow line path represents the C-terminal extension, and the green arrow line path indicates an additional fine-tuning process for C-terminal extension. **b** VAE-MH sequence encoding and decoding. **c** VAE-MH Pretrain and fine-tuning models for peptide extension.

of the encoding is decoded back to the original discrete representation. The reconstruction error is evaluated using the cross-entropy. Further details are presented in "Methods".

The second component is to use the Metropolis-Hastings (MH) algorithm to generate new peptide sequences by approximating $q(\mathbf{z}|c)$, where $\mathbf{z}$ is the encoding of an extension, $c$ is the variable corresponding to a desired molecular property (i.e., a PPI binder), and $q(.)$ is a probability distribution function (Fig. 2c). We chose the MH algorithm because it allows $q(.)$ not to be limited to any specific distribution form for a long vector $\mathbf{z}$. To use the MH algorithm, we need to know $q(\mathbf{z}|c)$ that is proportional to the product of $q(c|\mathbf{z})$ and $q(\mathbf{z})$. To obtain $q(c|\mathbf{z})$, we prepared a dynamic labeled dataset, in which each peptide is labeled as a potential PPI binder ($c$ = positive) or not a PPI binder ($c$ = negative). For $q(\mathbf{z})$, we used a Gaussian mixture model (GMM) to approximate the probability distribution of the extension encoding in all the training sequences. After obtaining $q(c|\mathbf{z})$ and $q(\mathbf{z})$, we utilized the MH algorithm to sample positive $\mathbf{z}$ from $q(\mathbf{z}|c=1)$, and then decoded it back to the original representation of a peptide extension. The details of the MH algorithm are presented in "Methods".

Peptide extensions are sampled in an iterative manner. In the first iteration, no target-specific binding data are available for both extensions. Thus, we used two datasets of potential PPI binders from PixelDB[41] and THPdb[42] that contain experimentally validated peptide binders to pretrain the peptide generation model. All the sequences in the two PPI datasets were labeled as positive (potential PPI binder), while all negative sequences (not potential PPI binder) were obtained from random sampling of the Uniprot database[40]. This model is referred to as the Pretrain model below. In the subsequent iterations, the labeled dataset was updated with the Rosetta FlexPepDock evaluation results of all peptide extensions generated in the preceding iterations. The top 10% scored peptides were considered as positive while the rest was taken as negative. Through this reinforcement learning-like strategy, our model, referred to as the Fine-tune model, was enabled to generate target-specific peptide extensions.

To reduce the variance of $q(c|\mathbf{z})$, we bootstrapped the labeled dataset to create four labeled datasets. A machine-learning model was trained on each of the four datasets to obtain four sets of model parameters. Using the four trained models, the classification of $\mathbf{z}$ based on the input $c$ was performed four times. The overall $q(c|\mathbf{z})$ was finally obtained as the average of the four results of $q(c|\mathbf{z})$. For better estimation of $q(c|\mathbf{z})$, we tested several machine-learning models and found that the support vector classifier (SVC) worked the best for N-terminus extension and the extreme gradient boosting (XGBoost) achieved the best performance in C-terminus extension. The results of the various classifier models are summarized in Supplementary Table 2.

## Rosetta FlexPepDock evaluation of N-terminal peptide extensions targeting β-catenin

We first tested the VAE-MH Pretrain model that was trained with a general dataset on N-terminal extension of the parent peptide. In total, 100 peptide extensions were generated for each length of 2–5 residues; the total length of the extended peptides thus ranges from 17 to 20 residues. Binding evaluation with Rosetta FlexPepDock shows that the probability distributions of I_sc and rmsALL_if shift to the left-hand side while the distributions of I_bsa shift to the right-hand side with the increasing peptide length (Fig. 3a), indicating that the N-terminal extension tends to boost the peptide–protein binding with only small perturbation to the binding pose of the parent peptide.

Previous studies have demonstrated that N-terminally extended peptides could enhance antibody affinity or activity by a factor of up to 20 compared to the parent peptide[33,43]. However, the computational study by Sood and Baker that aimed to increase the affinity of a protein–peptide complex by designing N- or C-terminal extensions only led to modest affinity increases[7]. It was hypothesized that the

modest affinity improvement might stem from the highly polar extensions interacting with the solvent rather than with the protein. To shed light on the N-terminal extension mechanism, we plot I_sc versus I_bsa for all the extended peptides in Fig. 3b. I_sc shows a better correlation ($r^2 = 0.4$) with I_bsa than with rmsALL_if ($r^2 = 0.2$). Furthermore, rmsALL_if shows a skewed probability distribution centered around −0.8 Å, indicating that the extensions do not perturb the parent peptide binding pose. These results suggest that the improved binding arises primarily from a better engagement of the N-terminally extended residues with the protein, rather than a rearrangement of the peptide-β-catenin-binding conformation (Fig. 3b).

Most peptide design relies on mutation to simultaneously optimize the sequence and structure for affinity maturation[7]. However, local permutation of amino acids limits search of a global energy minimum. Our method represents the first deep learning-based model for peptide extension, which reduces the search space by learning from examples of good PPIs and then conducts a more focused search instead of a random search. To compare the performance of our VAE-MH model with that of a traditional computational design approach, we plot the probability distributions of I_sc, I_bsa and rmsALL_if from our design along with those from the Rosetta Design in Fig. 3c. Our VAE-MH Pretrain model outperformed Rosetta Design as evidenced by the significantly decreased I_sc and increased I_bsa scores. Not surprisingly, rmsALL_if showed similar distribution patterns for the two models. Taken together, these data substantiate the effectiveness of our VAE-MH Pretrain model in designing N-terminal extensions with increased protein-binding affinity.

## Downward hierarchical selection of N-terminal peptide extensions targeting β-catenin

We next used a hierarchical strategy to select N-terminally extended peptides for experimental testing. Peptide ranking was performed sequentially with two binding affinity calculation methods, Rosetta FlexPepDock[44] followed by MM/GBSA[45]. Both methods provide an efficient way to evaluate the binding energy of a peptide–protein complex based on a known binding mode. Rosetta FlexPepDock performs structural refinement (minimization) of a peptide–protein complex using a Monte Carlo method and then estimates the binding energy for each minimized conformation using a Rosetta scoring function, while MM/GBSA samples an ensemble of peptide–protein complex conformations through molecular dynamics (MD) simulations and the binding energy is computed as a sum of gas phase energy (MM), electrostatic solvation energy (GB), and nonpolar solvation energy (SA). The MM/GBSA method capable of capturing both conformational flexibility and solvent effect is more accurate than the scoring function-based FlexPepDock, but remains computationally demanding. Therefore, the combination of the two methods in a hierarchical manner is expected to improve our prediction accuracy and speed.

To test the performance of Rosetta FlexPepDock and MM/GBSA in predicting the binding energies of our β-catenin-binding peptides, we collected 14 β-catenin inhibitory peptides that have been experimentally assayed, and computed the Rosetta and MM/GBSA-binding energies for ten peptides comprising only natural amino acids (Supplementary Table 1). The computed binding energies versus the experimental $IC_{50}$ values are plotted in Supplementary Fig. 2a. Rosetta FlexPepdock can only distinguish good binders from poor binders since most peptides have their binding scores greater than −30 REU and only 4 peptides have their binding scores below −35 REU. On the other hand, the MM/GBSA results correlate reasonably well with the experimental values. These results support our two-stage selection strategy−Rosetta FlexPepdock is used for initial screening, which is followed by more accurate rank-ordering with MM/GBSA. Specifically, we first ranked the 300 N-terminally extended peptides (100 peptides for each of the 3, 4, and 5 residue extension lengths) generated by the

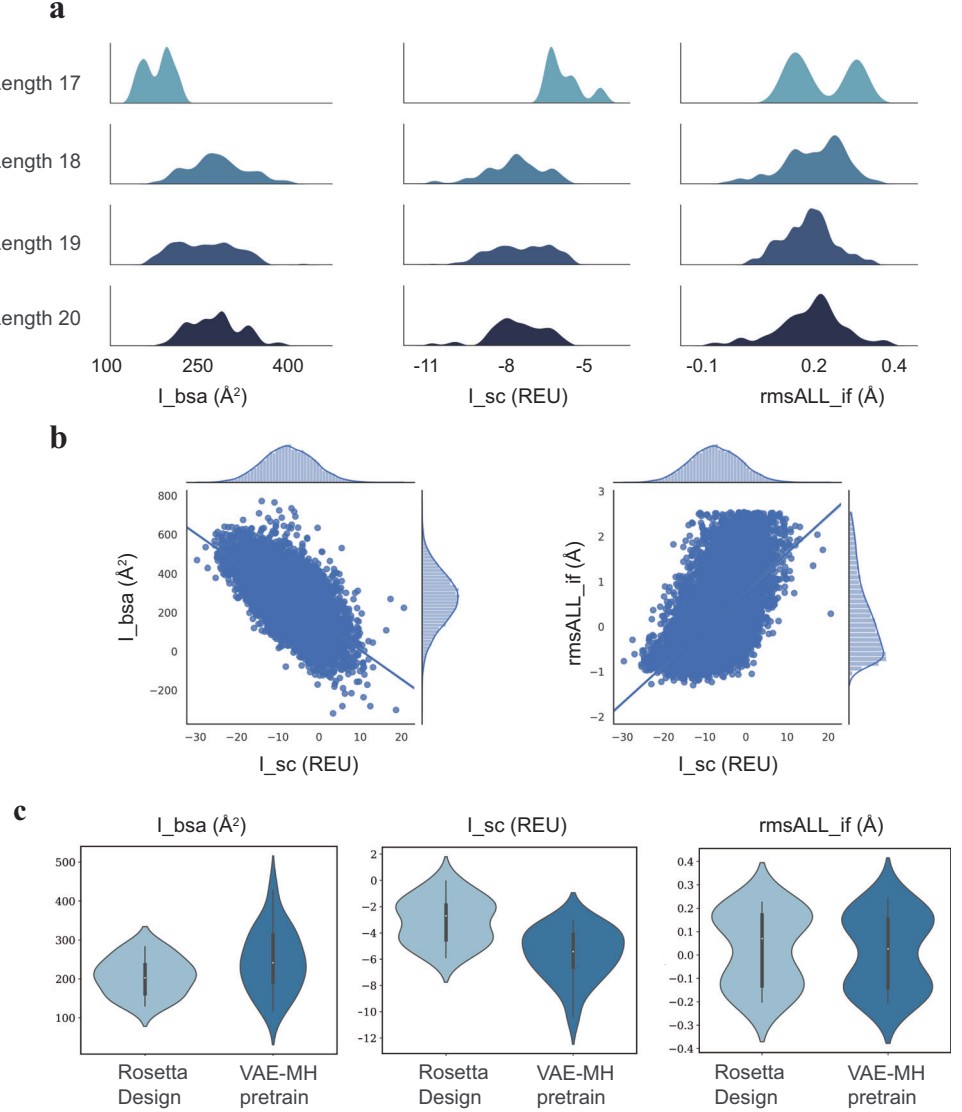

**Fig. 3 | Rosetta FlexPepDock evaluation of the VAE-MH generated N-terminal peptide extensions. a** Kernel density distribution plots for interface score (I_sc), interface binding solvent accessible area (I_bsa) and root-mean square deviations of interface backbones (rmsALL_if) of N-terminally extended peptides (17–20 residues). **b** Correlation plots of rmsALL_if vs. I_sc and I_bsa vs. I_sc for the N-terminally extended peptides, with each point representing a peptide. For each peptide, 100 conformations are sampled for each peptide–protein complex, and their FlexPepDock scores are averaged. **c** Violin plots of rmsALL_if, I_sc and I_bsa for N-terminal peptide extensions generated by the VAE-MH Pretrain model in comparison with Rosetta Design. FlexPepDock scores were computed as the difference score between the extended peptide and the parent peptide. Sample sizes in (**a**, **b**) are $n = 300$. Coefficients of determination in (**b**) for I_bsa vs. I_sc are $r^2 = 0.4$, and for

I_bsa Vs. rmsALL_if are $R^2 = 0.2$. The minima, maxima, center (median) bounds of I_bsa are 130, 283, 203 for Rosetta Design, and 118, 314, 241 for VAE-MH pretrain. Whiskers represent the interquartile range Q1 = 162 and Q3 = 237 for Rosetta Design and the interquartile range Q1 =192 and Q3 =314 for VAE-MH pretrain. The minima, maxima, center bounds of _sc are −5.8, −0.068, −2.7 for Rosetta Design, and −10, −3.1, −5.4 for VAE-MH pretrain. Whiskers represent the interquartile range Q1 =−4.5 and Q3 =−1.9 for Rosetta Design, and the interquartile range Q1 =−6.6 and Q3 =−4.1 for VAE-MH pretrain. The minima, maxima, center bounds of rmsALL_if are −0.20, 0.23, −0.071 for Rosetta Design, and −0.2, 0.24, 0.027 for VAE-MH pretrain. Whiskers represent the interquartile range Q1 = −0.13 and Q3 = −0.17 for Rosetta Design, and the interquartile range Q1 = −0.14 and Q3 = 0.15 for VAE-MH pretrain. Source data are provided as a Source Data file.

VAE-MH Pretrain model based on their collective Rosetta binding scores, and the 10 top-ranked peptides were then each subjected to binding free energy calculation with MM/GBSA. Table 1 summarizes the MM/GBSA results for the ten peptides. Compared to the parent peptides, the FlexPepDock scores of the top ten extended peptides are improved by 5–10 REU, and their MM/GBSA-binding energies improved by 4–8 kcal/mol. Interestingly, the Rosetta FlexPepDock results show little or no dependency on the peptide length, while the MM/GBSA results are slightly biased toward longer extensions.

To guide the selection of peptides for experimental testing, we computed the MM/GBSA-binding energies for all the 14 $\beta$-catenin inhibitory peptides in Supplementary Table 1. The MM/GBSA and IC$_{50}$

rankings show a good correlation $r^2$ of 0.6 (Supplementary Fig. 2b). Thus, the mean MM/GBSA-binding free energy of $43 \pm 3$ kcal/mol for the positive control group (with an IC$_{50} < 0.15\,\mu$M) was taken as a threshold filter to select peptides likely with improved potency. For robustness of the selection, the threshold was set to −40 kcal/mol that is one standard deviation higher than the mean value. Accordingly, two peptides NAL-2 and NAL-3 that have an MM/GBSA-binding energy greater than the threshold value were discarded (Table 1).

Besides the downward hierarchical selection strategy, we also explored alternative ways to optimize the VAE-MH-generated peptides for increased $\beta$-catenin-binding affinity. Since it is impractical to compute the gradient of binding affinity with respect to amino acid

**Table 1 | MM/GBSA-binding energy and experimental IC$_{50}$ results for the top 10 N-terminally extended β-catenin-binding peptides ranked by Rosetta FlexPepDock[*]**

| Peptide | Extended residues | Rosetta FlexPepDock (REU) | MM/GBSA (kcal/mol) | in vitro IC$_{50}$ (μM) |
|---|---|---|---|---|
| NAL-1 | RYSYPEDILDKHLQRVIL | −39.9 ± 6.0 | −46.2 ± 8.5 | Not tested |
| NAL-2 | LYDYPEDILDKHLQRVIL | −38.5 ± 6.1 | −33.9 ± 5.9 | Not tested |
| NAL-3 | WHSYPEDILDKHLQRVIL | −39.5 ± 6.4 | −35.9 ± 3.0 | Not tested |
| NAL-4 | SQRPYPEDILDKHLQRVIL | −38.6 ± 5.9 | −44.2 ± 8.2 | 0.17 |
| NAL-5 | IWWWYPEDILDKHLQRVIL | −39.3 ± 6.2 | −46.8 ± 6.7 | Not tested |
| NAL-6 | SGKVSYPEDILDKHLQRVIL | −37.5 ± 5.2 | −48.0 ± 8.9 | 0.10 |
| NAL-7 | RALRLYPEDILDKHLQRVIL | −38.2 ± 6.0 | −43.6 ± 8.5 | Not tested |
| NAL-8 | VYFWQYPEDILDKHLQRVIL | −39.0 ± 6.5 | −44.1 ± 5.4 | Not tested |
| NAL-9 | EGEKQYPEDILDKHLQRVIL | −38.2 ± 5.3 | −46.1 ± 8.1 | 0.084 |
| NAL-10 | AGSQPYPEDILDKHLQRVIL | −37.9 ± 6.3 | −42.2 ± 2.0 | >3 |

[*]Peptide extensions are underlined.

composition, we chose two evolutionary algorithms (EAs)—genetic algorithm (GA) and simulated annealing (SA). However, neither method yielded more potent peptide inhibitors. Compared to those obtained directly from VAE-MH (Supplementary Tables 3 and 4), peptides optimized by GA or SA show significantly higher (less favorable) MM/GBSA-binding energies (Supplementary Fig. 3). We suspect that the reason is twofold. First, there is a discrepancy between the Rosetta-style loss function used in the optimization and the molecular mechanics (MM) energy function used in MD simulation. In our optimization algorithms, the less accurate loss function cannot fully capture the conformational flexibility of the peptide–protein complexes, leading to the failure of the two EA-based methods. Second, unlike the VAE-MH model that samples in the encoded space, GA and SA operate directly in the amino acid sequence space. That means, peptide properties can change rapidly and irregularly, and even deviate significantly from the initial peptide sequence, thus losing the advantage of GA and SA in local optimization.

### Experimental validation and structural investigation of N-terminally extended β-catenin-binding peptides

Eight top-ranked peptides were prioritized for experimental testing from the combined Rosetta FlexPepDock and MM/GBSA evaluation (Fig. 4a, b). In addition, we inspected the polarity of these peptides and discarded four peptide extensions composed primarily of hydrophobic residues, which are more likely to cause non-specific binding. Finally, 4 peptide candidates were selected, synthesized, and assayed for their binding affinities for β-catenin through competitive fluorescence polarization (FP)-based binding experiments. Our experimental results (Table 1 and Supplementary Fig. 4) revealed that two out of the four designed peptides show improved potency over the parent peptide (IC$_{50}$ = 0.15 μM), with an IC$_{50}$ of 0.084 μM for the best peptide NAL-9 ("EGEKQ", Fig. 4c), which is approximately twofold better than the parent peptide. Only one of the designed peptides NAL-10 ("AGSQP") shows much worse performance than the parent peptide, with an unexpectedly high IC$_{50}$ value >3 μM. Inspection of the FlexPepDock poses indicates that the poor binding of the peptide can be attributed to the low helicity of the extension. The added N-terminal alanine and glycine residues do not form a stable secondary structure and appear to interact more extensively with the solvent rather than the protein (Fig. 4d). The best extension "EGEKQ" on the other hand comprises primarily charged residues glutamic acid and lysine, which tend to interact strongly with the protein (Fig. 4e).

To gain further insight into how the extended residues interact with β-catenin, we performed MD simulations of β-catenin bound with the four designed peptides, each for 500 ns. We analyzed the contact maps between the extended residues and the protein (Supplementary Fig. 5). The potent binders with an IC$_{50}$ < 0.15 μM all appear to make close contact with V208 and E209, which are however not involved in the interaction with the less potent peptide. Inspection of the binding poses of the best (NAL-9) and the worst (NAL-10) peptides reveals that R212 interacts favorably with the extended E1 and E3 of peptide NAL-9 (Fig. 4d), but forms an intramolecular salt-bridge with E209 and is not engaged in any interaction with peptide NAL-10 (Fig. 4e). Comparison of the MD results of the two peptide–protein complexes suggests that binding of negatively charged residues in the binding pocket dissociates R212 from E209 and releases E209 to the solvent. Interestingly, the resulted local conformational change could affect the solvent accessibility of a nearby residue C213. C213 appears to engage the third N-terminal residue of both peptides as if the extended helical turn is flipped to interact with the β-catenin-binding cleft (Fig. 4d, e). Thus, our computational results suggest a possibility of exploiting the solvent-exposed C213 for the development of potential covalent inhibitors targeting β-catenin.

### N-terminally extended β-catenin-binding peptides from library screening

Given the modest affinity improvement of the VAE-MH-derived peptides, we next explored the limit of the N-terminal extension strategy by chemically synthesizing and screening a combinatorial peptide library against β-catenin to see if more potent peptide inhibitors can be identified. We designed a one bead-one compound (OBOC) peptide library in the form of Ac-X$_1$X$_2$X$_3$X$_4$-YPEDILDKHLQRV-BBRM-resin, where each library member contained the core β-catenin-binding motif, YPED ILDK HLQRV (the underlined residues formed a lactam staple), an N-terminal extension of four random residues (X$_1$–X$_4$), and a C-terminal linker sequence (BBRM; B is β-alanine) for the purpose of library screening and hit identification. Each random position (X$_1$–X$_4$) was constructed with a set of 29 amino acids, including 7 proteinogenic amino acids (Gly, Ala, Ser, Ile, Asp, Gln, and His), 12 α-D-amino acids (D-Ala, D-Pro, D-Val, D-Thr, D-Leu, D-Asn, D-Lys, D-Glu, D-Phe, D-Arg, D-Tyr, and D-Trp), and 9 non-proteinogenic amino acids amino acids (β-Ala, D-β-homoAla, L-homoproline (Pip), cis-2-aminocyclopentylcarboxylic acid (cis-Acp), aspartic acid α-tert-butyl ester (Isa), L-phenylglycine (Phg), D-2-naphthylalanine (D-Nal), L-4-fluorophenylalanine (Fpa), L-norleucine (Nle), and L-ornithine (Orn)). The inclusion of noncanonical amino acids was intended to increase the structural diversity as well as the proteolytic stability of the library members.

Note that the parent motif, YPEDILDKHLQRV was modified from Peptide 9 by removing the two C-terminal residues to facilitate the identification of any library member of improved affinity caused by N-terminal extension. The shortened Peptide 11 (Supplementary Table 1) binds β-catenin with ~tenfold lower affinity than Peptide 9. The library has a theoretical diversity of ~710,000 unique compounds and

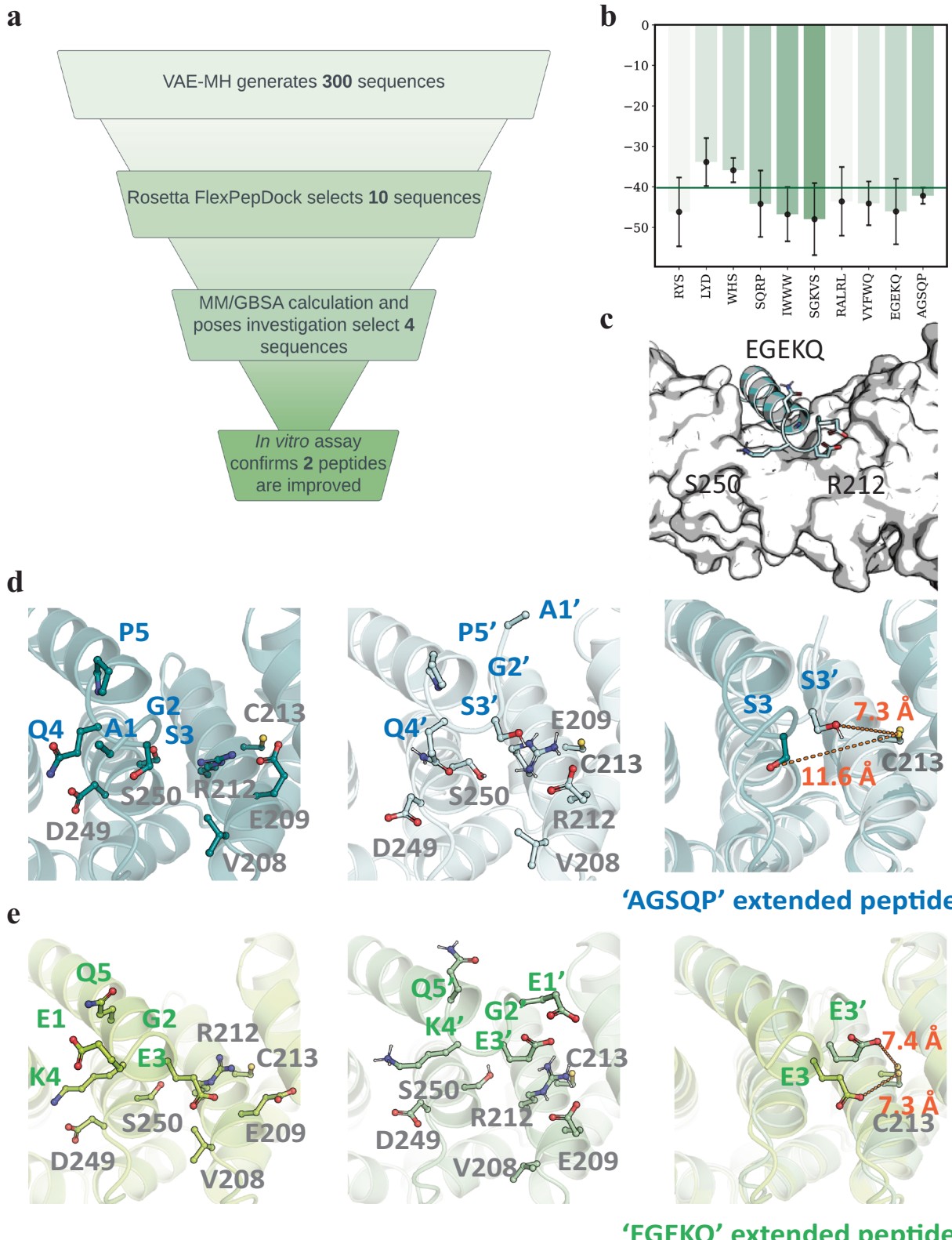

**Fig. 4 | MD simulations of β-catenin bound with the four experimentally tested peptides. a** Workflow of our N-terminal extension design process. **b** MM/GBSA results of the top 10 Rosetta FlexPepDock-scored peptide extensions. Error bars represent standard deviations (from left to right in kcal/mol: 8.5, 5.9, 3, 8.2, 6.7, 8.9, 8.5, 5.4, 8.1, 2). The green horizontal line denotes the threshold cutoff of −40 kcal/mol. **c** Ending pose of the best N-terminally extended peptide NAL-9 bound to β-catenin after 500 ns of MD simulation. **d** Initial and final structures along with their superimposition of the best peptide NAL-9 bound to β-catenin during the MD simulation. **e** Initial and final structures along with their superimposition of the worst peptide NAL-10 bound to β-catenin during the MD simulation. Extension residues and β-catenin residues crucial to binding are labeled and represented in stick. In (**b**), whiskers indicate standard deviations for triplet measurements. Source data are provided as a Source Data file.

was synthesized on 1 g of TentaGel S NH$_2$ resin (90 μm, 100 pmol peptide/bead). Approximately 300 mg of the library (~8.6 × 10$^5$ beads) was subjected to two rounds of screening for binding to β-catenin as detailed in "Methods" (magnetic screening followed by enzyme-linked assay). Surprisingly, during the enzyme-linked assay, most of the library beads developed similar, intense turquoise color during the 20-min incubation time, suggesting that these library members bind to β-catenin with similar affinities. We collected 25 most intensely colored beads (which were slightly more colored than the remaining beads), released the peptides by CNBr cleavage (after Met), and determined the peptide sequences by MALDI/TOF mass spectrometry. Out of the 25 beads, 24 produced unambiguously sequences (Supplementary Table 5). Inspection of the sequences showed that β-catenin prefers a cationic residue (e.g., D-Arg and Orn) at position X$_4$ but no obvious preference at positions X$_1$–X$_3$ (Supplementary Fig. 6).

We selected five representative sequences for resynthesis and binding analysis, based on the above preferences as well as the inclusion of noncanonical amino acids for proteolytic stability (Supplementary Table 6). It turns out that four out of the five sequences (L2, L12, L17, L19) bind β-catenin with a slightly lower affinity than the core sequence (by 1.5–2-fold) while the remaining one (L20) has a similar affinity to the parent peptide (Supplementary Fig. 4). These results suggest that the N-terminal extension may not be an effective strategy to significantly improve peptide potency as the OBOC random extension library failed to discover any peptides that are more potent than the parent peptide. On the other hand, these unsuccessful library screening results also demonstrate the power of our combined AI and structure-based design in identifying promising peptide binders from a very large pool of peptides, as two out of the four computationally designed peptides showed improved (likely the maximal possible) affinity for N-terminally extended peptides.

## Iteratively fine-tuning VAE-MH for the design of C-terminally extended β-catenin-binding peptides

Given the limitation of N-terminal extension, we shifted our focus to C-terminal extension to design significantly more potent β-catenin binders than the parent peptide. In peptides isolated from phage display libraries[33], only two C-terminal positions did not have a strong preference for a specific residue (positions 14 and 15 of the peptide $^1$YPEDILDKHLQRV$^{14}$I$^{15}$L). Most StAx peptides have Trp and Arg at the last two C-terminal positions, of which the Trp residue at position 14 was thought to increase the binding affinity by engaging recognition pockets of the surface of β-catenin while the terminal Arg residue was disordered in the crystal structure, and did not contact β-catenin[32]. In addition, our in-house data indicate substitution of the C-terminal "IL" with bulkier unnatural amino acids, such as TertLeucine, 3-Bta, and 2-Nal, increases the binding affinity by fivefold to the 0.03 μM range while removal of both residues reduces the binding affinity by tenfold (Supplementary Table 1). Therefore, we extended the parent peptide "YPEDILDKHLQRVIL" by adding up to seven amino acid residues to the C-terminus of the peptide to probe if we can discover extended peptides with significantly improved binding affinity (Fig. 5a). Given the importance of the two terminal residues Ile and Leu for binding, we removed these two residues to yield a short-formed parent peptide "YPEDILDKHLQRV", and extended this truncated parent peptide by adding up to nine amino acid residues to the C-terminus to see if our computational design can recover the affinity loss of the two terminal residues.

Our Pretrain model trained on a large peptide database has learned the sequence representation rule to represent peptide sequences in a continuous latent space. However, valid peptide sequences are not necessarily peptide inhibitors that bind specifically to β-catenin. Thus, we need to feed the model with target-specific data to obtain probability distributions of peptide sequences that are conditioned to a specific protein target, β-catenin. Unfortunately,

specific peptide-β-catenin-binding data are scarce. To tackle this issue, we iteratively fine-tuned our model with the Rosetta FlexPepDock scores of the already generated peptide sequences in two cycles (Fig. 5a).

Since Rosetta FlexPepDock scoring is used as the first layer to filter out peptides for the second layer evaluation with MM/GBSA, only those top-scored regions are most relevant. To test if this strategy can enrich the best peptide candidates, we first used the three FlexPep-Dock scores I_bsa, I_sc, and rmsALL_if to rank-order the peptides from the best to the worst with rank scores ranging from 0 to 1. Only those top-ranked peptides were used in the fine-tune cycles. We trained four models: Pretrain trained on a general dataset, Fine-tune1 fine-tuned with the Pretrain results, Fine-tune2 and Fine-tune3 both fine-tuned with the Pretrain and Fine-tune1 results. The difference between Fine-tune2 and Fine-tune3 is that the latter was trained on 600 additional Pretrain peptides. Random conserved mutations of Ile and Leu at positions 14 and 15, such as Val, Ala, Ile, and Leu were introduced to these peptides to test the importance of these two hydrophobic residues for potent peptide-β-catenin interaction.

We computed the rank scores for peptides generated by each of the Pretrain, Fine-tune1, Fine-tune2, and Fine-tune3 models and then combined all peptides together to compute their overall rank scores. For each peptide, the rank difference is taken as each of the four individual rank scores (for Pretrain, Fine-tune1, Fine-tune2, and Fine-tune3 models, respectively) minus the overall rank score. The higher the rank difference, the better the corresponding model can enrich top-ranked peptides. As shown in Fig. 5a, both YPEDILDKHLQRVIL- and YPEDILDKHLQRV-based extensions show a clear trend of increasing rank differences during fine-tuning, demonstrating that our fine-tuning strategy can enrich "good" peptide binders and is thus more likely to generate peptides with higher affinity for β-catenin. Interestingly, no distinct shift to higher rank differences is observed after Fine-tune1 (Fig. 5a). Nevertheless, for YPEDILDKHLQRVIL-based extension, the Fine-tune3 model did improve its overall rank compared to Fine-tune2 and Fine-tune1.

We plot the probability distributions of the FlexPepDock scores in Fig. 5b, c for peptides generated by the three Fine-tune models and the Pretrain model. Compared to the Pretrain model, the probability distributions of I_sc and rmsALL_if shift to the left-hand side while the distributions of I_bsa to the right-hand side for all three fine-tune modes, indicating that the C-terminal extension could boost the peptide–protein interaction for both parent peptides when the VAE-MH model was fed with more specific data. We further set three cutoff values of -6 REU, 250 Å$^2$ and −0.2 Å for I_sc, I_bsa, and rmsALL_if, respectively, and plot in scatter plots the peptides that have the FlexPepDock scores above the corresponding cutoff values (Supplementary Fig. 7). As illustrated in the pie charts, all three Fine-tune models show a clear enrichment of top-ranked peptides compared to the Pretrain model.

A total 162 peptides with their FlexPepDock scores above the cutoff values were subjected to MM/GBSA calculation (Fig. 6a). The MM/GBSA results show that the YPEDILDKHLQRV-based extensions have a wider binding energy distribution than the YPEDILDKHLQRVIL-based extensions (Fig. 6b). Ten top-ranked peptide extensions for each of the two parent peptides were chosen for experimental synthesis and binding assay (Fig. 6c). Sequence alignment of the top 10 YPEDILDKHLQRV-based extensions indicates that our pipeline can recapitulate the binding favorable hydrophobic property of the C-terminus of the parent peptide (position 14 and 15) to propose hydrophobic/aromatic residues, such as Trp, Phe and Tyr for these positions. Surprisingly, Asp and Glu are also frequently observed in these positions (Fig. 6e). Why are negatively charged Asp and Glu residues favored in these positions and often followed by aromatic residues? Inspection of the crystal structures of the β-catenin bound with TCF, LEF1 and TCF4[46–48] reveals that all the peptide motifs bound

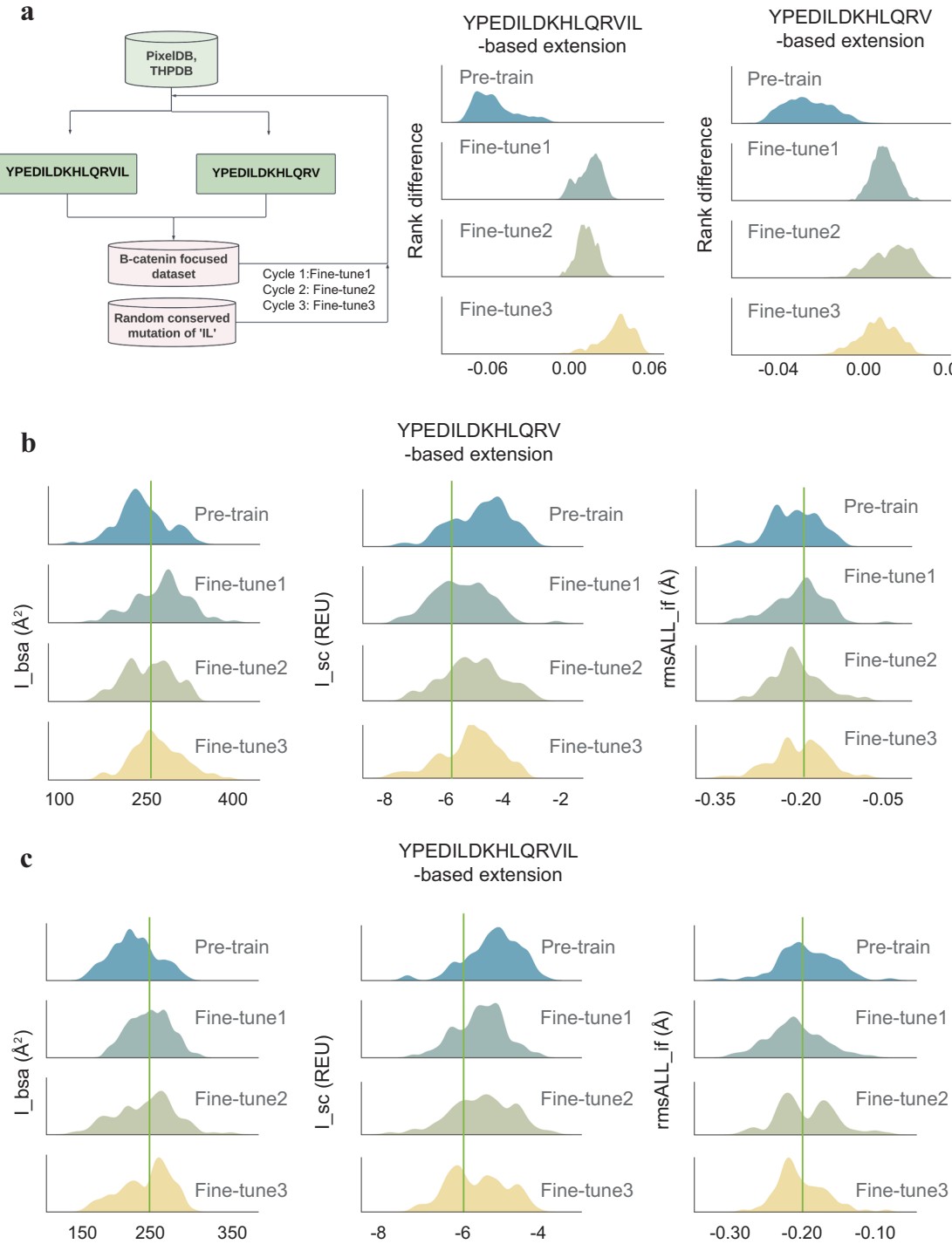

**Fig. 5 | Rosetta FlexPepDock evaluation of C-terminal peptide extensions generated by the three fine-tune models in comparison with the Pretrain model. a** Workflow of the three fine-tune models for C-terminal extension, and comparisons of enrichment results analyzed by rank difference distributions for C-terminal extensions generated by various models. **b** Distributions of the three Rosetta FlexPepDock binding metric scores evolving during the fine-tuning process for YPEDILDKHLQRV-based extension. **c** Distributions of the three Rosetta

FlexPepDock binding metric scores evolving during the fine-tuning process for YPEDILDKHLQRVIL-based extension. Only the top 10% Rosetta FlexPepDock scores are plotted. The green line represents a cutoff for selecting peptides subjected to MM/GBSA evaluation (rmsALL_if < −0.2 Å; I_sc < −6 REU; I_bsa > 250 Å²). Sample sizes in (a) for YPEDILDKHLQRVIL-based extension and YPEDILDKHLQRV-based extension are $n = 1840$ and $n = 2480$, respectively. Sample sizes in (**b**, **c**) are $n = 400$. Source data are provided as a Source Data file.

to the binding site 3 of β-catenin are enriched in Asp and Glu, and these negatively charged residues appear to form very favorable interaction with Q302 and R376 of β-catenin (Fig. 6d). Since all of the TCF sequences were included in our training dataset, it is thus not surprising that our VAE-MH model learned to conditionally generate negatively charged residues followed by aromatic residues in these positions. Unlike the YPEDILDKHLQRV-based extensions, Asp or Glu does not occur frequently in the YPEDILDKHLQRVIL-based extensions, instead, aromatic residues Trp, Phe and Tyr are obviously more abundant (Fig. 6e, f). We suspect that the difference may arise from the fact that the two additional hydrophobic residues increase the probability of the occurrence of hydrophobic residues in the following

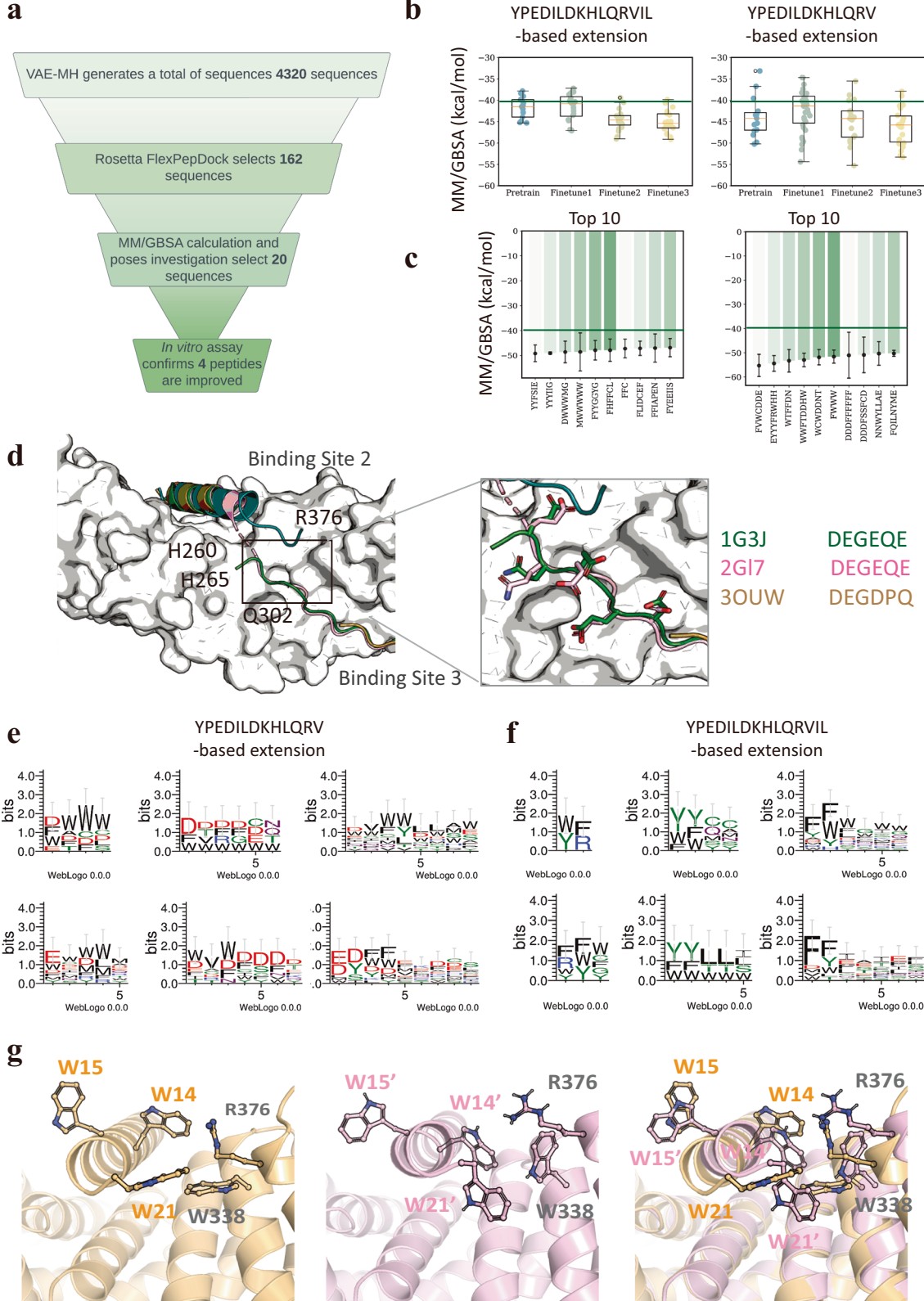

positions as their generation is not only learned from the training data but also depends on the preceding sequences.

Finally, eight peptide candidates were selected, synthesized, and assayed for their binding affinities for β-catenin through competitive fluorescence polarization (FP)-based binding experiments (Table 2 and Supplementary Fig. 4). Four of our designed peptides show improved potency over the parent peptide, with the best peptide CAL-2 having

an IC$_{50}$ of 0.010 μM, which represents a 15-fold improvement over the truncated parent peptide. Three of our designed peptides show slightly decreased binding affinity than the parent peptide, with their IC$_{50}$ values in the 0.3 μM range. Interestingly, all these three peptides are derived from the full-length parent peptide and contain Ile-Leu at positions 14 and 15, while the best extension CAL-2 ([14]WWFTDDH[21]W) has two Trp residues at positions 14 and 15. Interestingly, when

**Fig. 6 | Computational and experimental evaluation of our designed C-terminal peptide extensions. a** Workflow of our C-terminal extension design process. **b** MM/GBSA energy distributions for top peptides generated by the four VAE-MH models. **c** MM/GBSA energies of top ten peptides for each of the two C-terminal extensions. Error bars represent standard deviations. **d** Crystal structures of TCF-bound β-catenin (PDBid: 1G3J, 2GL7, and 3OUW). **e** Sequence alignment of selected peptide extensions (with MM/GBSA-binding energy below −40 kcal/mol) for YPEDILDKHLQRV-based extension (4–9 residues). **f** Sequence alignment of selected peptide extensions (with MM/GBSA-binding energy below −40 kcal/mol) for YPEDILDKHLQRVIL-based extension (2–7 residues). **g** Initial and final structures along with their superimposition of the best peptide CAL-2 bound to β-catenin during the MD simulation. Extension residues and β-catenin residues crucial for binding are labeled and represented in stick. Sample sizes in (**b**) for YPEDILDKHLQRVIL-based extension and YPEDILDKHLQRV-based extension are $n = 45$ and $n = 54$, respectively. In the left panel of (**b**), boxes bound the interquartile range of (Q1 = −45.3, median = −41.5, Q3 = −37.8) for Pretrain, (Q1 = −43.8, median = −40.8, Q3 = −39.2) for Fine-tune1, (Q1 = −44.6, median = −43.6, Q3 = −44.6) for Fine-tune2, and (Q1 = −46.5, median = −45.4, Q3 = −43.2) for Fine-tune3; whiskers represent the most extreme values of (minima = −45.3, maxima = −37.8) for Pretrain, (minima = −47.1, maxima = −37.2) for Fine-tune1, (minima = −49.1, maxima = −39.4) for Fine-tune2, and (minima = −49.2, maxima = −39.9) for Fine-tune3. In the right panel of (**b**), boxes bound the interquartile range of (Q1=−47.0, median = −44.2, Q3 = −42.9) for Pretrain, (Q1 = −45.4, median=−41.4, Q3 = −39.1) for Fine-tune1, (Q1 = −48.6, median=−44.3, Q3 = −42.5) for Fine-tune2, and (Q1 = −49.8, median = −45.8, Q3 = −43.7) for Fine-tune3; whiskers represent the most extreme values of (minima = −50.2, maxima = −33.2) for Pretrain, (minima = −54.4, maxima = −34.7) for Fine-tune1, (minima = −55.3, maxima = −35.5) for Fine-tune2, and (minima = −53.3, maxima = −37.9) for Fine-tune3. In (**c**), whiskers indicate standard deviations for triplet measurements. Source data are provided as a Source Data file.

transferring our designed sequences to experimental lab for synthesis and binding assays, a mistake was made on the CAL-7 sequence. Upon realizing the mistake, we decided to continue to test this "mistaken" peptide and use its result as a control. As expected, the peptide fails to show any affinity improvement and has a similar affinity ($IC_{50} = 1.4$ μM) to the parent peptide (Table 2). Given the small difference between CAL-7 (Fine-tune3) and CAL-7 (error), this result highlights the ability of our computational approach in designing highly specific β-catenin binders.

To gain further insight into how the extended residues, especially the two Trp residues in positions 14 and 15, improve the β-catenin affinity, we investigated the binding pose of the best peptide CAL-2 with β-catenin (Fig. 6g). W14 is found to engage with R376 through a pi-cation interaction, and W21 forms a π−π stacking interaction with W338, a hot spot residue on β-catenin. On the other hand, W15 does not interact with β-catenin and instead points away to the solvent. During the MD simulation, the side chains of all these residues rearrange themselves significantly. W21 moves slightly away from W338, but the engagement is maintained throughout the simulation. W14 forms a new interaction with W338 while interacting simultaneously with R376. These results provide an explanation why the C-terminal extensions starting with aromatic residues followed by negatively charged residues can increase the β-catenin-binding affinity of the parent peptide.

### Iteratively fine-tuning VAE-MH for the design of NEMO-binding peptides

To demonstrate its broad applicability, we further applied our computational strategy to a more challenging and less explored system: the NF-κB essential modulator (NEMO). NF-κB is a transcription factor responsible for activating genes involved in inflammation, immune response, and cell survival[49]. Abnormal NF-κB signaling is implicated in various autoimmune diseases and cancers. Under physiological conditions, NF-κB signaling is regulated through the interaction between the inhibitor of κB (IκB)-Kinase (IKK) and NF-κB essential modulator (NEMO)[50–52]. Binding of IKK to NEMO activates IKK, leading to the phosphorylation and degradation of IκB, resulting in increased NF-κB activity. One strategy to target elevated NF-κB activity is to inhibit the NEMO-IKK interaction, thereby increasing IκB activity and reducing NF-κB activity[53]. We aimed to design N-terminal extensions for a known NEMO-binding peptide to enhance its potency against the NEMO-IKK interaction.

We employed the 11-residue NEMO-binding domain (NBD) of IKKβ ($^{735}$TALDWSWLQT$^{745}$E in IKKβ numbering) as a template for extension (Fig. 7a). The parent peptide, NBD, which includes the conserved hexapeptide LDWSWL, exhibits weak binding to NEMO, with an $IC_{50} \gg 100$ μM. The NEMO/IKKβ peptide complex presents a structure of an asymmetrical, parallel four-helix bundle. Each NEMO molecule resembles a crescent-shaped α-helix, and two of these helices form a

dimer by packing head-to-head[53]. The dimeric NEMO features a flat slit, forming two broad and extensive IKK-binding pockets. NBD peptides are wedged between the interfaces of the NEMO dimer at the C-terminus, with the IKK-binding surfaces extending from the N to C termini. NBD occupies only the C-terminus pocket, leaving a substantial cavity in the middle of the helix bundle for N-terminal extension (Fig. 7a).

Consequently, we applied our VAE-MH workflow to design N-terminally extended NBD peptides, aiming to enhance their binding affinity for NEMO (Fig. 7b). However, designing peptide inhibitors for NEMO presents a substantial challenge for two primary reasons. First, both NEMO and IKK exhibit considerable conformational flexibility, as demonstrated by the crystal structure. The lack of a regular secondary structure in the majority of NBD makes it difficult to predict accurately the interaction strengths between designed peptides and NEMO. Second, the flat and extensive binding surface of NEMO/IKK lacks distinct pockets or grooves that could be targeted by peptide inhibitors, complicating the design of smaller-sized peptide inhibitors to disrupt this interaction.

In light of these challenges, we placed greater emphasis on sequence space sampling rather than extensive structural space refinement with MD simulations (Fig. 7b). We adopted the second computational pipeline used in the design of C-terminally extended β-catenin inhibitors, which combined VAE-MH sampling with multiple fine-tuning cycles. In each cycle, we sampled random sequences and calculated their interface energies using PyRosetta with the Rosetta energy function[54]. The energy distribution from one cycle guided the sampling of the next batch of sequences in the subsequent cycle. During the fine-tuning cycles, the interface distributions shifted towards a more favorable direction (Fig. 7c).

Unlike the β-catenin system, we did not rely solely on the MM/GBSA method for ranking and prioritizing peptides for synthesis and assays. Instead, we scrutinized the interaction modes of the top 67 peptides generated from VAE-MH with NEMO. We also considered the polarity, diversity, and length of these peptides. Following MD simulation and structural inspection, we selected four peptide sequences with favorable binding poses and other properties for synthesis and binding assays. As summarized in Table 3, two of the tested peptides (NBD+2 and NBD+12) exhibited significantly improved binding relative to NBD, with $IC_{50}$ values of ~50 μM and ~75 μM, respectively. The other two peptides (NBD+1 and NBD+4) were only slightly more potent than NBD, with $IC_{50}$ values exceeding 100 μM (Supplementary Fig. 8).

The binding structures of our four designed peptides in NEMO are illustrated in Fig. 7d. Notably, none of these peptide extensions adopt a regular secondary structure. The binding site for the NBD extension is quite polar, consisting of residues K90, R87, Q86, and F82, favoring negatively charged residues. This explains why NBD+2, primarily composed of polar residues with the N-terminus capped by a Trp residue that stacks against F82, and NBD+12, comprising two Asp

**Table 2 | MM/GBSA and experimental IC$_{50}$ results of the eight C-terminally extended β-catenin-binding peptides[*]**

| Peptide | Peptide sequence | MM/GBSA (kcal/mol) | in vitro IC$_{50}$ (μM) |
|---|---|---|---|
| CAL-1(Fine-tune3) | YPEDILDKHLQRVWCWDDNT | −52.9 ± 3.2 | 0.078 ± 0.01 |
| CAL-2 (Fine-tune3) | YPEDILDKHLQRVWWFTDDHW | −52.9 ± 2.8 | 0.010 ± 0.006 |
| CAL-3 (Fine-tune1) | YPEDILDKHLQRVEYYYFRWHH | −54.4 ± 3.2 | 0.089 ± 0.013 |
| CAL-4 (Fine-tune2) | YPEDILDKHLQRVFVWCDDE | −55.3 ± 4.6 | 0.070 ± 0.021 |
| CAL-6 (Fine-tune2) | YPEDILDKHLQRVILYYYIIG | −49.1 ± 4.7 | 0.24 ± 0.038 |
| CAL-9 (Fine-tune3) | YPEDILDKHLQRVILYYFSIE | −49.2 ± 3.3 | 0.30 ± 0.037 |
| CAL-10 (Fine-tune2) | YPEDILDKHLQRVILFFC | −47.2 ± 3.8 | 0.20 ± 0.064 |
| CAL-7 (Fine-tune3) | YPEDILDKHLQRVILFHFFCL | −47.9 ± 4.5 | Not tested |
| CAL-7 (human error) | YPEDILDKHLQRVILFHFCIL | Not calculated | 1.4 ± 0.39 |

[*]Peptide extensions are underlined.

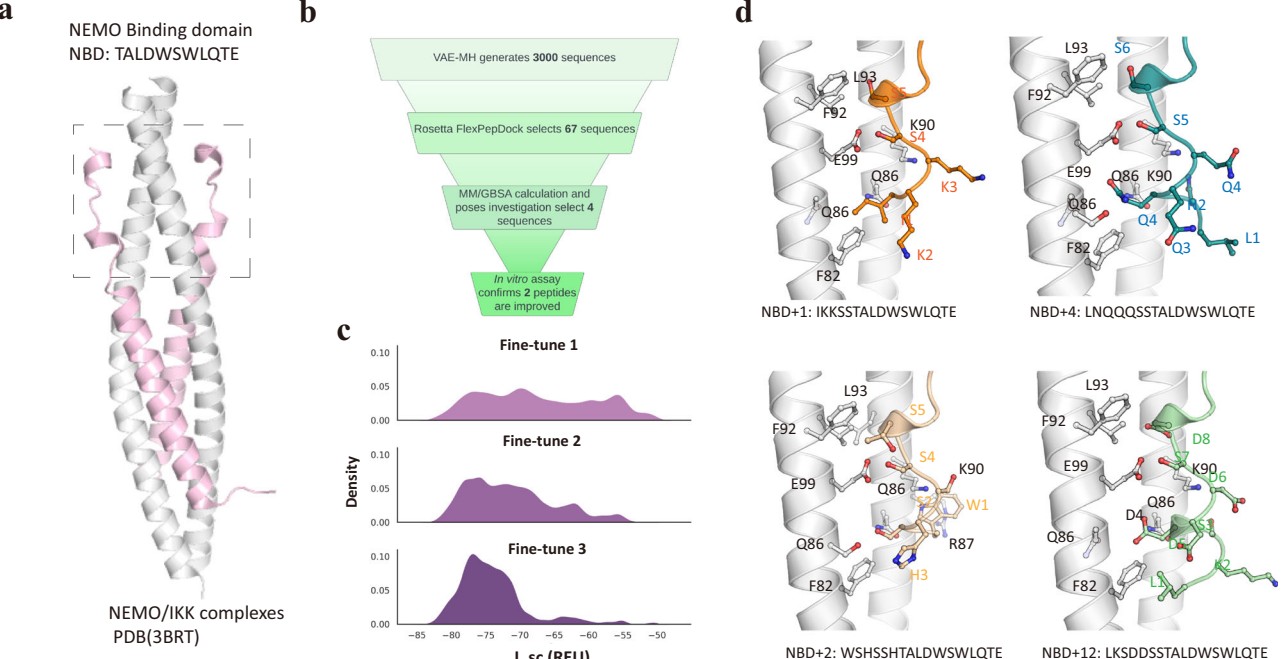

**Fig. 7 | Design of NEMO/IKK-binding peptide extensions. a** Crystal structure of the NEMO/IKK complex (PDBid: 3RBT[53]). **b** Workflow illustrating the peptide design process. **c** Distribution of interface energy (REU) throughout fine-tuning cycles.

**d** Visualization of peptide-NEMO binding poses for the four tested peptides. NEMO helices are depicted in gray, and peptide inhibitors are represented in distinct colors. Source data are provided as a Source Data file.

residues, exhibited stronger binding than the other two peptides. Remarkably, even without structural attention, our sequence-based VAE-MH model successfully captured the favorable interaction patterns and generated two peptide sequences with improved NEMO-binding compared to the parent peptide.

## Discussion

Deep generative models have achieved remarkable success in natural language processing, enabling the modeling of complex distributions in real-word audio and text data[16,18]. They have also proven to be a powerful tool for generating novel chemical and biological structures with desired properties[55,56]. Yet, to our knowledge, this strategy has not been extensively explored in the design of high-affinity peptide inhibitors targeting specific proteins. We introduce two integrated computational pipelines that combine deep learning, peptide docking, and molecular dynamics (MD) simulation to enhance the binding affinity of a parent peptide for β-catenin and NEMO, respectively, through the design of N- or C-terminal extensions.

Our first computational pipeline relies on a hierarchical downward selection strategy to rank-order the peptide extensions

generated by VAE-MH. This is necessary because the VAE-MH Pretrain model is not trained on the β-catenin specific binding data, so two-step evaluation with Rosetta FlexPepDock and MM/GBSA is leveraged to select peptides that have a high affinity for β-catenin. We employed this workflow to design N-terminal extensions. Two of the four designed peptides exhibited enhanced binding affinity for β-catenin, whereas the improvement was modest (e.g., twofold). Why was the affinity increase so modest? In the best N-terminal extension NAL-9, two of the five extended residues are negatively charged, complementary to the highly polar surface of the β-catenin-binding cleft. Consequently, both the extension and the protein likely face competitive interaction from the solvent, attenuating the peptide−protein binding. This finding is further supported by our library screening data, in which none of the N-terminally extended peptides showed any affinity improvement relative to the parent peptide. These results suggest that the N-terminal extension may not be an effective way to design peptides with significantly enhanced potency, probably due to its limited design space.

In our second computational pipeline, the VAE-MH model was iteratively fine-tuned to generate β-catenin-binding C-terminal

**Table 3 | MM/GBSA and experimental IC$_{50}$ results of the four tested NEMO inhibitors***

| Peptide | Peptide sequence | MM/GBSA (kcal/mol) | in vitro IC$_{50}$ (μM) |
|---|---|---|---|
| NBD+1 (Fine-tune3) | IKKSSTALDWSWLQTE | −19.2 ± 1.5 | >100 |
| NBD+2 (Fine-tune3) | WSHSSHTALDWSWLQTE | −17.9 ± 1.1 | 50 |
| NBD+4 (Fine-tune3) | LNQQSSTALDWSWLQTE | −14.9 ± 1.2 | >100 |
| NBD+12 (Fine-tune3) | LKSDDSSTALDWSWLQTE | −16.3 ± 1.1 | 75 |

*Peptide extensions are underlined.

extensions. Unlike the N-terminal extension, we observed a significant improvement in binding affinity with the C-terminal extension. Out of the eight tested C-terminal extensions, four exhibited enhanced binding affinity for β-catenin, with the best one, CAL-2, showing a remarkable 15-fold improvement over the parent peptide. Encouraged by this superior performance, we extended the application of the second computational pipeline to design N-terminal extensions targeting NEMO. As anticipated, two out of the four designed peptides demonstrated substantially enhanced binding affinity for NEMO compared to the parent peptide, NBD. These findings underscore the effectiveness of our transfer learning strategy in the design of target-specific peptide inhibitors. Our iterative fine-tuning VAE-MH model not only learns sequence representation within a large and general dataset, but also captures target-specific features within a small and target (i.e., β-catenin, NEMO)-focused dataset. While transfer learning has been utilized to design small molecule drugs and antimicrobial peptides to address data scarcity issue[57,58], its potential in the design of small molecules or peptides targeting specific proteins has remained largely unexplored, primarily due to the limited availability of target-specific data.

Overall, our computational models were able to generate hundreds of potential peptide sequences from an enormous residue sequence space (i.e., $20^5$–$20^9$). These sequences were subsequently subjected to peptide docking and MD simulations, resulting in the prioritization of a much reduced number (tens) of peptide candidates for experimental evaluation. Notably, half of the designed β-catenin inhibitory peptides, 6 out of 12, exhibited improved binding affinity compared to the parent peptide. Particularly remarkable was the performance of the C-terminally extended peptide CAL-2, which bound to β-catenin with an IC$_{50}$ of 0.010 ± 0.06 μM. This represents a 15-fold improvement over the parent peptide (0.15 ± 0.04 μM) and a threefold improvement over the most potent binder (0.037 ± 0.02 μM) in our training dataset. These results are particularly promising given that our designed peptides are composed solely of natural amino acids, in contrast to the utilization of unnatural amino acids and peptide stapling, which are commonly used for affinity maturation[59]. Furthermore, for the NEMO system, two out of the four designed peptide extensions demonstrated significantly enhanced binding to NEMO. This outcome is noteworthy for two reasons. First, the parent peptide, NBD, is unstructured and exhibits weak binding with an IC$_{50}$ ≫ 100 μM. Second, the NEMO/IKKβ complex is characterized by an elongated, loosely packed four-helix bundle, and NEMO and NBD do not make extensive contact at the dimer interface. Taken together, these findings underscore the effectiveness of our integrated computational pipelines in the design of potent peptidyl inhibitors tailored for specific protein targets.

Peptides hold a tremendous potential for targeting large and flat protein-protein interactions (PPIs)[60]. More than 80 therapeutic peptides are on global market and more than 100 peptides are in the various stages of preclinical or clinical studies[61]. As such, peptides have become a unique drug class for treating a wide range of diseases, including cancer, infectious diseases, diabetes, and gastrointestinal diseases[62]. Given this, robust approaches that leverage recent advances in deep learning are urgently needed to accelerate the design of target-specific peptide inhibitors. To this end, our computational pipeline that combines deep learning-based VAE-MH sampling with biophysics-based Rosetta docking and MD simulation provides a promising way to enable the design of peptide extensions with significantly improved binding affinity for a specific protein target. Besides peptide extension, our computational pipelines will also be useful for the de novo design of peptides where a well-defined peptide template exists for the protein of interest.

## Methods

### Data preparation

We created two datasets; one is labeled, and the other is unlabeled. We hypothesized that the co-crystallized peptides in protein–peptide complexes possess hidden features to learn for the design of novel protein-binding peptides. Thus, we considered these peptides as potential PPI binders and labeled them as positive. These "positive" peptides come from two main sources. The first is THPdb, which contains FDA-approved peptide drugs[42]. The original size of the dataset is 188. However, most of the peptides have length greater than 100. We randomly sliced these peptides into 80K peptide fragments of length up to 50 and considered these peptide fragments as potential PPI binders. The second source is PixelDB[63]. This dataset contains 1966 protein–peptide complex structures. We computed the interface energy $\Delta G$ using the Rosetta Interface module[64] to annotate these protein–peptide complexes. We kept the peptides whose $\Delta G$ is < −35 Rosetta Energy Unit (REU) and labeled them as potential PPI binders, which yielded 290 potential peptides. Since the lengths of all these 290 peptides are within 50, we did not slice this dataset. The rest of peptides were discarded instead of being labeled as negative. We denoted this dataset as PixelDB. The "negative" samples come from randomly sampling the Uniprot protein sequence database. In Uniprot, some sequences are "reviewed", meaning that the information on the corresponding proteins has been extracted from literature, and the rest is "unreviewed", meaning that the corresponding sequences were only computationally analyzed. We combined both reviewed and unreviewed sequences to create a unified Uniprot dataset. After filtering out peptides longer than 50 amino acids, we obtained a dataset of around 95K sequences. Since almost none of these peptides are known to exist in a peptide–protein complex structure, we considered the whole Uniprot dataset as negative samples. We randomly sampled 80% of the positive and negative sequences to form the training samples. The rest 20% is the test dataset. From the training samples, we randomly sampled 80% of them four times to form four labeled training datasets. The unlabeled dataset contains protein sequences from Uniprot, PixelDB and sliced THPdb. This dataset serves for training peptide sequence encoding. We remove all the sequences whose length is less than 17 and greater than 50. This reduces the size of our final unlabeled dataset to around 160K.

### Sequence representation

In many NLP tasks, a word is embedded into high-dimensional vectors for a meaningful representation[65]. We used word-embedding techniques similar to those used in NLP to represent peptide sequences. First, we split a peptide sequence into a base peptide (the central 15

residues) and an extension (the rest N- or C-terminal residues). Then we tokenize the two parts. We create a vocabulary of size 24. The vocabulary contains 24 numbers from 0 to 23, where 0–3 represent the start, the end, an unknown residue, and a blank space padding of the sequence, respectively, and 4–23 each represent one of the 20 natural amino acids. We map each part onto these number representations. For the base peptide, the length is fixed to 15. For the extension, we limit our sequence length to 35. Any sequences shorter than 35 will be padded with number 3 until the total length reaches 35. Second, we embed each of these numbers in their one-hot representation into a high-dimensional continuous vector space as the final representation of the peptide sequences. For example, for a tokenized extension of length $N$ and embedding size $M$, the representation will be $N \times M$ dimensional. This feature transformation process is done using a linear layer that is a part of our VAE network. Finally, the representation will be learned through the VAE training.

## Latent space encoding

The model components for the base peptide encoding and the peptide extension encoding are the same. The only difference is that the input for the base peptide model has a fixed length while the input for the peptide extension model has a varying length. Thus, the following description of the VAE model applies to both models.

In the encoding task, a gated recurrent unit (GRU) based variational autoencoder (VAE) is used, which involves an encoder E and a decoder D. E encodes a sequence $\mathbf{x}$ into a latent space vector $\mathbf{z}$. D decodes this latent space vector $\mathbf{z}$ and outputs $\tilde{\mathbf{x}}$ such that it is as similar to $\mathbf{x}$ as possible. Thus, the whole autoencoder is optimized based on a reconstruction error. VAE makes the autoencoder a generative model by learning the encoder D such that $D_{KL}(q_D(\mathbf{z}|\mathbf{x})\|p(\mathbf{z}|\mathbf{x}))$ is minimized, where $q_D(\mathbf{z}|\mathbf{x})$ is the distribution of $\mathbf{z}$ produced by the encoder D and $p(\mathbf{z}|\mathbf{x})$ is the true distribution of $\mathbf{z}$. Such an objective function cannot be optimized directly. However, minimizing this objective is the same as minimizing the negative evidence lower bound (ELBO)[66].

$$-\text{ELBO} = -E_{q_D(z|x)} \log p(x|z) + D_{KL}(q_D(z|x) \| p(z)) \quad (1)$$

Equation (1) is used as a loss function for learning our VAE model. The first term is the reconstruction error. We use cross-entropy (CE) to measure the error since our final output $\mathbf{x}$ is discrete. The second term is the distribution difference between the generated latent space vector $\mathbf{z}$ and some known prior of $\mathbf{z}$. In a typical setup, the prior $p(\mathbf{z})$ is set to be a unit Gaussian $N(0, 1)$ for each dimension. Thus, the final expression of our loss function is shown in Eq. (2),

$$\text{Loss}(\mathbf{x}, \tilde{\mathbf{x}}) = -\sum_{l=1}^{L}\sum_{v=1}^{V} x_{lv} \log(p(\tilde{x}_{lv})) + \sum_{k=1}^{K} q_D(z_k|\mathbf{x}) \log\left(\frac{q_D(z_k|\mathbf{x})}{p(z_k)}\right) \quad (2)$$

where,

$L$ is the length of the sequence and $V$ is the size of the tokens,

$x_{lv}$ is the value of the $v$th dimension of the $l$th token. Note here, each $\mathbf{x_l}$ is represented as a one-hot vector.

$p(\tilde{x}_{lv})$ is the probability that the $v$th dimension of the $l$th token is one.

$q_D(z_k|\mathbf{x}) = N(\mu_k, \sigma_k)$ is the normal distribution of the $k$th dimension of the encoding and the mean and variance are outputs of the encoder.

$p(z_k) = N(0, 1)$ is a one dimensional unit Gaussian distribution.

Since the input of the VAE model is $N \times M$ sequential data, we use a GRU network to process the data. The GRU network is a type of recurrent neural network (RNN) that is specially designed to process order-dependent data. It is a frequently used architecture in text processing, but faces significant challenges in processing long sequential data due to the vanishing gradient problem[67]. GRU

addresses this issue by adding gate mechanisms[68]. Compared to another frequently used long-short-term memory (LSTM) model, GRU has fewer parameters and, is thus computationally lighter. In our model, the encoder receives embeddings as inputs and uses the GRU network to process the embeddings. The initial hidden state is set to 0. The output of the GRU network is connected to two linear layers. One layer outputs the mean estimation of the encodings. The other layer outputs the variance of the encodings. The decoder of the network receives the mean estimation of the encodings and the embedding of the sequences as the inputs, which are processed by another GRU network. The mean of the encodings is treated as the initial hidden state of the GRU. A linear layer is connected to the output of the GRU. The result of the linear layer is $N \times 24$ dimensional, indicating the probability of the occurrence of 24 individual vocabularies in each position of the protein sequence. The ground truth sequences are represented using $N \times 24$ dimensions, and one-hot vector is used for the second dimension. Thus, the cross-entropy loss is calculated using the output of the decoder and the ground truth in this representation.

## Conditional sequence generation

The generation of desired peptide sequences is done by using the Metropolis Hasting (MH) algorithm[69]. To generate a peptide extension that is a potential PPI binder given the base peptide, we need a conditional distribution $q(\mathbf{z}|c)$, where $c$ is the label, $\mathbf{z}$ is the latent space encoding of the extension. However, the distribution form of $q(\mathbf{z}|c)$ can be arbitrary and the dimension of $\mathbf{z}$ needs to be high enough to reduce the reconstruction error in VAE. Thus, we use the MH algorithm, a Markov Chain Monte Carlo method known for high-dimensional sampling, to sample $\mathbf{z}$ from an unknown distribution $q(\mathbf{z}|c)$. This is achieved via the following observation. From the Bayes rule, $q(\mathbf{z}|c)$ can be written as,

$$q(z|c) = \frac{q(c|z)q(z)}{q(c)} \quad (3)$$

In Eq. (3), $c$ is fixed. Thus, $q(c)$ is constant independent of $\mathbf{z}$. As a result, to calculate the acceptance probability in the MH algorithm, the two terms that need to be computed are $q(c|\mathbf{z})$ and $q(\mathbf{z})$. Although VAE regulates $z$ to a unit distribution, the distribution $q(\mathbf{z})$ is not accurately represented by a unit Gaussian[21]. Thus, we approximate $q(\mathbf{z})$ using a Gaussian mixture model.

For $q(c|\mathbf{z})$, a support vector classifier (SVC) with a fivefold cross-validation and bootstrap strategy is used. For each of the four labeled training datasets, we fit SVC to obtain $q_n(c|\mathbf{z})$, where $n = 1, 2, 3, 4$. Then, $q(c|\mathbf{z}) = \frac{\sum_{n=1}^{4} q_n(c|\mathbf{z})}{4}$. We measure the algorithm performance on the unified test dataset. In Supplementary Table 2, we compare the different algorithms for estimating $q(c|\mathbf{z})$ in different iterations. The comparison shows that SVC with bootstrap yields the best result.

Details of the potential binder sampling process are given in Supplementary Methods 1 and 2. We consider a Gaussian distribution as the proposal distribution for $q(\mathbf{z}_i^{t+1}|\mathbf{z}_i^t)$. We use 500 burn-in iterations to initialize the MH sampling chain. Since Gaussian is symmetric, the proposal distribution is not involved in our acceptance rate calculation. To ensure computational stability, we compute the acceptance rate in a log scale. After a desired encoding is sampled, we decode it to a peptide extension sequence via our extension decoder. The decoding process is completed in a recursive fashion: $sequence_i = decoder(sequence_0, sequence_1, …, sequence_{i-1})$, where $sequence_i$ stands for the sequence token at the $i$th position of the sequence. $sequence_0$ is set to 0, which is the token to signal the start of the sequence. This process is recursive because we do not know the input sequence at the beginning. In the N-terminal extension, we only require peptide extensions of 2–5 residues. Thus, we loop over the MH algorithm to sample potential binders of a length up to 5 that do not

belong to the peptide extensions in the labeled dataset until 100 peptide extensions are obtained for each length. Finally, we concatenate the extensions with the base peptide to form complete peptides.

## Rosetta FlexPepDock

Rosetta FlexPepDock[11] was employed to estimate the binding energy of a peptide–protein complex structure with the Rosetta scoring function using the following four metrics: interface energy (I_sc), peptide score (pep_sc), root-mean-square of interface atoms (rmsALL_if), and buried surface area of the interface (I_bsa) that have been widely used for the evaluation of protein-protein interface energies[70]. I_sc is calculated by summing over the energies contributed by the interface residues of the peptide and the protein. The lower the I_sc and pep_sc values, the stronger the binding. I_bsa is calculated by subtracting the sum of the solvent accessible surface area (SASA) of each monomer from that of the complex. A larger I_bsa indicates a stronger peptide–protein interaction. RmsALL_if is the root-mean-square-deviation (RMSD) between an output model and the reference structure, so a smaller rmsALL_if value usually indicates better binding of the peptide with the protein. For each input peptide–protein complex, the interface energies are computed for the top 50 conformations sampled with Rosetta FlexPepDock[11], and their mean values are reported as the binding energies for the complex. Finally, we define a combined binding score as $\frac{pep\_sc \cdot I\_sc \cdot I\_bsa}{rmsALL\_if}$ to rank the VAE-MH-generated peptides. We also define an overall ranking score as the sum of the individual ranks for pep_sc, I_sc, I_bsa, and rmsALL_if.

## MM/GBSA-binding free energy calculation

Each of the top-ranked peptides based on their Rosetta FlexPepDock combined scores was subjected to MD simulation and MM/GBSA-binding free energy calculation. All MD simulations were conducted with the CHARMM27 force field using Gromacs 5.0[71,72]. MD simulations were started from the lowest REU conformations of the corresponding peptide–protein complexes sampled by Rosetta FlexPepDock. All peptide–protein complexes were solvated in a TIP$_3$P water box with 12 Å padding on each side of the box and were neutralized with the appropriate number of sodium and chloride ions. The resulting systems were first minimized with the steepest descent algorithm for 5000 steps, and then by the conjugated gradient algorithm for 10,000 steps. After minimization, the systems were heated to 310K and equilibrated for 1 ns in the NPT ensemble at a pressure of 1 atm using the Parrinello-Rahman pressure coupling[73]. The Particle Mesh Ewald (PME) method[74] with a cutoff distance of 14 Å was employed to handle the long-range electrostatic interactions, and a cutoff distance of 10 Å was also used for the truncation of the Lennard–Jones potentials. All bonds involving a hydrogen atom were constrained by the LINCS algorithm[75]. Finally, three independent production simulations were carried out each for 10 ns for each peptide–protein complex. In total, 1000 snapshots were evenly taken from each trajectory and then subjected to the endpoint binding energy calculation using GMX_MMPBSA[29]. The mean and standard deviation of the MM/GBSA-binding free energy were calculated over the three simulation runs. The modified Generalized Born model GB-OBC1[76] was used. The salt concentration was 0.15 M. The internal dielectric constant was 2 and the external dielectric constant was 80. The molecular surface area was calculated with a probe radius of 1.4 Å using the Linear Combination of Pairwise Overlaps (LCPO) algorithm[77]. The surface tension was set to 0.0072 kcal/(mol · Å$^2$) for the nonpolar contribution.

## β-Catenin peptide library synthesis

The peptide library was synthesized on 1.0 g of TentaGel S NH$_2$ resin (90 μm diameter, 0.26 mmol/g; Supplementary Fig. 9). Each coupling reaction was performed at RT using 5 equiv of Fmoc-amino acids and HATU/HOBt/DIPEA (5/5/10 equiv) for 1 h unless otherwise mentioned.

The N-terminal Fmoc were removed using 20% piperidine in DMF. A linker sequence (β-Ala-β-Ala-Arg-Met) was first synthesized followed by the common peptide region YPEDILDKHLQRV using the above conditions. Next, the variable region (X$_1$–X$_4$) was synthesized using the split-and-pool synthesis method[78]. In brief, the resin was divided into 29 equal portions by volume and each portion was transferred into a separate reaction vessel. To each vessel, 4.5 equiv of a different Fmoc-amino acid was added along with HATU/HOBt/DIPEA (5/5/10 equiv). Capping agents CD$_3$CO$_2$H, CH$_3$CD$_2$CO$_2$H, and/or CH$_3$CO$_2$H were added into each coupling reaction to generate 10% chain termination (for the synthesis of X$_1$–X$_4$ positions only) to facilitate sequence determination by mass spectrometry[79,80]. For amino acids with unique residual masses, 0.5 equiv of CD$_3$CO$_2$H was used, whereas 0.25 equiv of CH$_3$CO$_2$H and 0.25 equiv of CD$_3$CO$_2$H were included into the coupling reactions for D-Ala, D-Leu, D-Lys, Orn, Acp, Isa and 0.25 equiv of CH$_3$CD$_2$CO$_2$H and 0.25 equiv of CD$_3$CO$_2$H were added to the reactions for βAla and Nle. After the addition of the X$_4$ position, all the resin portions were pooled, exhaustively washed with DMF, and treated with 20% piperidine for Fmoc removal. The split-and-pool process was repeated to couple residues X$_1$-X$_3$. After the addition of the X$_1$ residue, the N-terminal Fmoc group was removed, and the N-terminus was acetylated by the treatment with 10 equiv of acetic anhydride. For peptide stapling, the Mtt and O-2-PhiPr protecting groups on Lys and Asp residues, respectively, were selectively deprotected using 2% TFA/1% TIPS in DCM (5 min × 6 times). The resulting Asp and Lys side chains were crosslinked by the treatment of PyBOP/HOBt/DIPEA (5/5/10 equiv) for 1.5 h twice at RT. The peptides were deprotected with reagent K for 3 h, washed thoroughly with DCM/DMF, and stored at − 20 °C in DMF.

## β-Catenin peptide library screening

For the first round of screening (magnetic sorting), 300 mg of the library resin was washed extensively with H$_2$O and HBST-gelatin buffer (30 mM HEPES, 150 mM NaCl, pH 7.4, 0.05% Tw-20, 0.1% gelatin, 2 mM DTT and 3% BSA). Biotinylated β-catenin protein was added to the HBST-gelatin buffer at a concentration of 100 nM and incubated with the resin for 6 h at 4 °C. The solution was drained and washed three times with the HBST-gelatin buffer to remove any excess biotinylated protein. In total, 50 μL of SA-coated Dynabeads (Invitrogen) in the HBST-gelatin buffer was added to the resin and incubated with gentle mixing for 20 min at 4 °C. After washing with the HBST-gelatin buffer to remove unbound magnetic particles, the resin was slowly added along the side of a 15-mL Falcon tube set in a magnetic particle concentrator (TA Dynal MPC-1). Positive beads (beads with Dynabeads on their surface) were attracted to the concentrator wall while the negative beads settled at the bottom of the tube. The positive beads were transferred to a 0.8-mL Bio-Rad column and washed with the HBST-gelatin buffer. For the second round of screening, 10 nM biotinylated β-catenin in the HBST-gelatin buffer was mixed with the positive beads from above and incubated for 6 h at 4 °C. Next, the resin was drained gently and incubated with 1 μg/mL streptavidin-alkaline phosphatase (SA-AP) in the HBST-gelatin buffer at 4 °C for 10 min. After washing with the HBST-gelatin buffer (3 × ) and staining buffer (30 mM Tris, pH 8.5, 100 nM NaCl, 5 mM MgCl$_2$, 20 μM ZnCl$_2$) (3 × ), the beads were transferred to a Petri dish using 1 mL of staining buffer. To the Petri dish, 100 μL of 5 mg/mL of 5-bromo-4-chloro-3-indolyl phosphate (BCIP) was added, and the mixture was incubated at RT on a rotary shaker. An intense turquoise color developed on positive beads in 20 min and the staining reaction was quenched by the addition of 1 mL of 1 M HCl. The most intensely colored beads (25 beads) were manually isolated with a micropipette under a dissecting microscope. The sequences were determined using MALDI-TOF mass spectrometry[79].

## Individual peptide synthesis and labeling

β-Catenin peptides were manually synthesized by SPPS on Rink amide resin by using Fmoc chemistry and 2-(7-aza1H-benzotriazole-1-yl)-

1,1,3,3-tetramethyluronium hexafluorophosphate (HATU) as the coupling agent. Coupling reactions typically involved Fmoc-amino acids (5 equiv.), HATU (5 equiv.) and diisopropylethylamine (DIPEA; 10 equiv.), and were carried out at room temperature (RT) for 45 min. Peptides were cleaved off the resin and deprotected by treatment with 92.5% TFA, 2.5% water, 2.5% triisopropylsilane (TIPS), and 2.5% 1,3-dimethoxybenzene at RT for 3 h. The solvents were removed by flowing a stream of $N_2$ over the solution, and the residue was triturated three times with cold Et$_2$O. The crude peptides were purified by reversed-phase HPLC equipped with a Waters XBridge C18 column, which was eluted with linear gradients of acetonitrile (containing 0.05% TFA) in ddH$_2$O (containing 0.05% TFA). For peptides containing a side-chain lactam cross-link, Lys(Mtt) and Asp(O-2-PhiPr) were incorporated at their designated positions during manual SPPS using the previously indicated coupling reagents. Following completion of the linear sequence, the N-terminal Fmoc group was removed and acylated with Ac$_2$O (10 equiv.) and DIPEA (10 equiv.) in DCM for 10 min twice. Acid-labile side-chain protecting groups were removed by incubating the resin with 2% TFA and 1% TIPS in DCM three times for 5 min. Lactam formation was performed using PyBOP (5 equiv.) and DIPEA (5 equiv.) in 1:1 (v/v) DMF/DCM for 2 h followed by overnight incubation. Peptides were washed, and any remaining amine was acylated using Ac$_2$O (10 equiv.) and DIPEA (10 equiv.) in DCM (2 × 10 min). For peptides containing a fluorescent label, precursor peptides were first synthesized and purified. Approximately 1 mg of lyophilized peptide was incubated with 5 equiv. of an activated fluorescent labeling reagent (e.g., FITC or 5(6)-carboxyfluorescein succinimidyl ester) and 5 equiv. of DIPEA in 150 μL of 1:1 (v/v) DMF/150 mM sodium bicarbonate (pH 8.5) for 2 h. The reaction was quenched by TFA, the labeled peptides were purified again by HPLC, and their authenticity was confirmed by MALDI-TOF mass spectrometry.

NEMO peptides were synthesized on a CEM Liberty Blue microwave-assisted peptide synthesizer at 25 μmol scale with CEM ProTide Rink amide resin using standard Fmoc chemistry. Each coupling reaction consisted of 8 equivalents of diisopropylcarbodiimide, 4 equivalents Oxyma pure, and 4 equivalents Fmoc-amino acid and were carried out at 90 °C for 4 min, except for arginine (which was coupled twice at 90 °C for 4 min), cysteine and histidine (the latter two were coupled once at 50 °C for 10 min to reduce epimerization). Deprotection of Fmoc was performed immediately at room temperature following the coupling of aspartate residues to reduce aspartimide formation. For biotinylation of IKK$\beta$, the peptide, while still resin-bound, was treated with 4 equivalents of NHS-Biotin and 10 equivalents of DIPEA for 1 h at RT. Peptides were deprotected and cleaved from resin with a 90:2.5:2.5:2.5:2.5 solution of TFA, DODT, H2O, thioanisole, and TIPS for 3 h at room temperature, followed by precipitation in cold ether. Crude peptides were dissolved in a minimal volume of DMF, diluted in 50:50 water and acetonitrile, and purified by reversed-phase HPLC. Peptide purity (>95%) and authenticity were confirmed on a Waters Acquity UPLC system connected to an SQD2 ESI-MS. The peptide concentration was determined from their absorbance at 280 nm. The chromatographic and mass spectrometric data of all peptides synthesized in this work are provided in Supplementary Figs. 10 and 11.

### GST-NEMO expression and purification
*E. Coli* BL21 (DE3) cells were transformed with a pGEX4T3-NEMO(1-196) expression vector and grown at 37 °C in Luria-Bertani broth supplemented with 75 mgL ampicillin. Expression was induced with 0.25 mM isopropyl-$\beta$-D-1-thiogalactopyranoside (IPTG) when cells reached an OD$_{600}$ of 0.6, and were allowed to proceed for 5 h at 30 °C. The cells were pelleted by centrifugation for 30 min at 4230 × *g*. The cell pellet was resuspended in 50 mL of lysis buffer (25 mM Tris, 150 mM NaCl, 5% glycerol, pH 7.4) supplemented with two tablets of cOmplete™, EDTA-free Protease Inhibitor Cocktail (Roche), 15 mg of lysozyme, and 25 mg of phenylmethylsulfonyl fluoride dissolved in 50 μL of DMSO. The cell lysate was briefly sonicated and clarified by centrifugation for 30 min at 26900 × *g*. The supernatant was then loaded onto 2 mL of glutathione-agarose resin (Pierce). The resin was washed with 20 column volumes of wash buffer (25 mM Tris, 150 mM NaCl, 5% glycerol, pH 7.4). Protein was eluted with 5-column volumes of elution buffer (25 mM Tris, 150 mM NaCl, 5% glycerol, 10 mM glutathione, pH 8) and exchanged into the wash buffer by using a VivaSpin 10-kDa MWCO centrifugal unit. Protein concentration was determined by Bradford's assay and absorbance at 280 nm.

### Peptide inhibitor binding assays
The in vitro IC$_{50}$ of the predicted peptides against $\beta$-catenin was measured through a fluorescence polarization (FP)-based binding assay. Expression and purification of GST-$\beta$-catenin has been reported previously[37]. FAM-labeled probe peptide (10 nM) was incubated with 50 nM GST-$\beta$-catenin in 20 mM Tris, 300 mM NaCl, pH 8.8, 0.01% Triton-X-100 for 1 h. Serial dilutions of a competitor peptide were prepared in 20 mM Tris, 300 mM NaCl, pH 8.8, and 0.01% Triton-X-100. After 1 h, aliquots of the equilibrated probe peptide-$\beta$-catenin solution were added to serially diluted peptide solutions and incubated for 1 h at RT. Samples were transferred into black-on-black 384-well nonbinding microplates (Greiner), and FP was measured using a Tecan M1000 Infinite plate reader. The data were analyzed using GraphPad Prism v. 8.0 and normalized to FP values corresponding to the fully bound/unbound probe. All raw data from FP-based competition assays are provided in Supplementary Fig. 4.

The binding affinities of the designed peptides for NEMO were measured by a competitive homogeneous time-resolved fluorescence (HTRF) assay. GST-NEMO (20 nM), biotin-IKK$\beta_{KK/RR}$ (aa701-745, 50 nM), streptavidin labeled with d-2 acceptor (Cisbio; 1.3 μg/mL), and anti-GST monoclonal antibody labeled with Tb donor (Cisbio; 0.06 μg/mL) were mixed in PBS (pH 7.4) containing 1 mM TCEP and 0.01% Triton X-100. The anti-GST monoclonal antibody (Revvity Health Sciences, catalog number: 61GSTTLA, lot number: 16RA) has been validated by Revvity QA using an HTRF assay with GST-biotin and compared to a reference batch. Varying concentrations of peptide (0–100 μM) were added in a white, shallow-well 384-well plate (Grenier) to a total volume of 20 μL. HTRF signal was measured on a Tecan Infinite M1000 plate reader using the manufacturer's recommended setting for Terbium HTRF. HTRF signal was normalized and analyzed using GraphPad Prism 6 software. All experiments were performed in triplicates ($n = 3$). All raw data from the HTRF assays are provided in Supplementary Fig. 8.

### Reporting summary
Further information on research design is available in the Nature Portfolio Reporting Summary linked to this article.

## Data availability
The peptide sequence datasets used for training VAE models and PPI binder classifiers in the VAE-MH process in this study are accessible on Zenodo at https://doi.org/10.5281/zenodo.10587692, under the Attribution-NonCommercial (CC BY-NC) license. Source data are provided with this paper.

## Code availability
The code associated with this study has been published on Zenodo at https://doi.org/10.5281/zenodo.10587692, under the Attribution-NonCommercial (CC BY-NC) license. Our sample code can be executed through Code Ocean at https://codeocean.com/capsule/5785222/tree.

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

## Acknowledgements

Research in the Cheng lab was partially supported by The Ohio State University (OSU)'s Translational Data Analytics Institute (TDAI) Interdisciplinary Research Pilot Award. Work in the Pei lab was supported by NIH grant GM122459 and National Cancer Institute (NCI) grant CA234124. This research used the resources of the Ohio Supercomputer Center (OSC), the OSU's Campus Chemical Instrument Center (CCIC) supported by NIH P30CA016058 award, and the Bruker ultrafleXtreme MALDI supported by NIH 1S10RR025660-01A1 award.

## Author contributions

S.C. and T.L. designed AI models, performed simulations, designed peptides, and wrote the manuscript. R.B. and J.R. performed experimental validation and wrote the manuscript. S.W. organized the dataset. Y.L. organized the code. X.L. revised the manuscript. X.C. and D.P. conceived the idea and designed the study. X.C., L.B., and D.P. supervised the study and revised the manuscript. All authors read and approved the submission of the manuscript.

## Competing interests

The authors declare no competing interests.
