## [Peer Review File · Nature Communications]

Reviewers' Comments:

Reviewer #1:

Remarks to the Author:

The manuscript by Chen et al. reports the in silico affinity maturation of a previously reported stapled peptide ligand of the protein beta-catenin. This protein is a central oncogene in the Wnt signalling pathway and a long-standing target of drug discovery efforts. This is mainly due to the fact that only a few validated synthetic binders have been reported so far, and that they possess only low cellular uptake. In this respect, the affinity maturation of an existing ligand is of limited use to address these issues. However, if generally applicable this approach may be useful for other relevant target proteins or for the de novo design of inhibitors. For that reason, I see potential for publication in *Nat. Commun.* I do not have the expertise to evaluate the computational workflow and therefore focused on the introduction and experimental parts.

Overall, the manuscript makes a preliminary impression with central experimental data and information missing. See the following list for details:

The authors write: „All of the tested compounds, including PNU-74654, UU-T01, UU-T02, MSAB, HI-B1, and 4FNPC had K_d values in the micromolar range, indicating that none of these primary hits are sufficiently potent.“ For many of those compounds, binding to beta-catenin could not be reproduced (see, McCoy *J. Med. Chem.* 2022, 65, 7246).

Considering that the manuscript reports peptide ligands, it would be more appropriate to cite alternative confirmed peptide-inspired ligands of beta-catenin (e.g. Schneider, *Nat. Commun.* 2018, 9, 4396; Wendt, *Angew. Chem. Int. Ed.* 2021, 60, 13937; Blosser, *ACS Chem. Biol.* 2021, 16, 1518).

The authors write: “As a transcription factor, the β -catenin also features a significant degree of structural disorder.” This is not correct since i) beta-catenin is not a transcription factor but a transcriptional coactivator, and ii) structural disorder is not a prerequisite for a transcription factor (neither for a transcriptional coactivator). Also, the authors refer to a stapled peptide in the next sentence, which binds to the folded armadillo repeat domain of beta-catenin and not its flexible regions. This should be clarified.

The authors refer to an example in which a CPP was fused to the aforementioned stapled peptide and write that „both binding affinity and specificity ... were compromised” (reference 5). Here, the study should be mentioned that reports a cell permeable CPP-modified peptide that shows high affinity and selectivity for beta-catenin (Ditrich, *Cell Chem. Biol.* 2017, 24, 958).

Readability of Figure 1 would improve if consistent colour code was used (e.g. same color for beta-catenin). Also, it is not clear why the CPP modification is shown and not just the original crystal structure.

The SI should contain analytics of all synthesized peptides and plots of raw data that was used to calculate reported IC₅₀ values.

With respect to the in silico affinity maturation of peptide ligands the authors should describe the state of the art (here just one example: Rooklin, *J. Am. Chem. Soc.* 2017, 139, 15560).

The authors write: “...peptide 9 bound to β -catenin with an IC₅₀ value of 145 \pm 35” What is the unit of this value?

Readability of Table 1 would be improved if sequences were aligned and a monospaced font would be selected.

The authors should give a bit of background regarding one-bead-one compound libraries (e.g. Lam lab and many others).

Experimental peptide affinities have only been assessed using FP companion assays. The authors should perform an alternative assay (e.g. ITC) and also determine K_D values in a direct FP with

fluorescently labelled peptides.

The authors could test activity of their peptides in cell-based assays when using CPP-modified versions (as done by the study serving as the starting point).

Reviewer #2:

Remarks to the Author:

The paper by Chen et al. reports a generative algorithm for the generation of optimized peptides for binding to B-catenin. They used a known binder of B-catenin and used their machine learning method to extend the N-terminal of the peptide. Followed by Rosetta FlexPepDock and MD, they picked some peptides to experimentally test and showed that some of their peptides achieve 2x enhanced binding. They also showed that a random extended library does not perform as well.

Overall the reviewer thinks that there is not enough evidence that this method is in fact a powerful approach to creating better binders given the marginal improvement in binding with the need for so much computational work. Since the method only applied to one example of a helical binder that has been extended, the reviewer cannot see how it can be generalized to more complicated cases where such starting motif does not exist.

The reviewer also finds the flow of the paper a little confusing due to inconsistencies across.

Examples:

1. Some tables use $\times 10^2$ nM (which is a strange choice) and some use μ M
2. They mention their best AI-based peptide has 84 nM IC50. But the table shows $8.4 \times 10^2 = 840$ nM.
3. The parent peptide used for the comparison of synthetic library has different binding affinity than the one used for the computational one.
4. They refer to structural info in figure 3C-E but it seems like they meant 1C-E

The rationale behind why they did what they did and whether it truly holds promise is questionable to the reviewer. Here are some areas of question:

1. The best binder from AI method has an IC50 of 84 nM. It seems like their control set (Table 1) already includes better binders and many of them are obtained via substitutions in C-terminal.
2. The authors use a number of scores from Rosetta FlexPepDock to decide N-terminal extension works better than C-terminal extension. They also used these measures as selection criteria in other parts of their work. However, nowhere in the paper one can see a plot showing that these scores or their combination reflect experimental binding affinity (they have a plot for their MD method, 7A). Without that, the use of these metrics doesn't feel justified.
3. In figure 5, all the sampled AI models as well as the random model show better values than the control set except I_bsa. Given that there are some very good binders in the control set, the reviewer is curious why this is happening. It seems like perhaps the inclusion of the two outliers (peptides 12 and 13 with very high IC50) might have skewed the distribution. One would expect random extensions don't work better than controls overall unless any extension is better than no extension, which does not sound plausible.
4. Comparing the AI model with just random sequences seems unfair to the reviewer given that there are faster methods with which one can generate extensions. For example, the authors could have generated extended peptides using Rosetta and used Rosetta Design to assign sequences. That would be a better comparison in terms of state-of-the-art compared to random. Especially given that their model is much slower (due to the requirement of MD step)
5. The authors did a great job documenting their code. However, details of the AI model are not clear in the text. For example, it is unclear what the fixed size for the GRU unit is. If the unlabeled sequences are at 50 and only 15 is kept as the base, it means that 35 residues will be used for extension. However, 35 is really far from the true extension value used here, 2-5 AA.
6. The reviewer wonders why the authors did not pick a random subset of AI-generated sequences for experimental library synthesis to compare the performance. It also would be good to get either a visual or mathematical summary of the library screening.

7. How do the authors verify that their random UniProt sample does not include binders? Many proteins bind to other proteins for function.
8. The differences in Rosetta FlexPepDock scores do not seem significantly meaningful to the reviewer based on their experience. It's also unclear if p-value of 0.1 is calculated given the big error that MM/GBSA has with regards to predicting true binding affinities and whether that shows any significant difference between AI VAE2, GA and SA.

The figures can improve:

1. It's really hard to get any conclusion about figures 1C-E about structure (authors referred to those for justification of extension)
2. The reviewer is curious why the convention of using violin plots was not used in fig. 5. The lines need to be thicker for that figure.
3. In figure 9, are the starting point the predicted docked peptides from FlexPepDock? Also the reviewer thinks "Structural Analysis" can be a little misleading of a title for 2.8 as it suggests that there are real structures. You only know these are models by reading the figure caption.

We appreciate the reviewers' constructive comments and suggestions. Presented below are our point-to-point responses to the reviewers' comments.

Point-to-point response to Reviewer #1:

The manuscript by Chen et al. reports the in silico affinity maturation of a previously reported stapled peptide ligand of the protein beta-catenin. This protein is a central oncogene in the Wnt signalling pathway and a long-standing target of drug discovery efforts. This is mainly due to the fact that only a few validated synthetic binders have been reported so far, and that they possess only low cellular uptake. In this respect, the affinity maturation of an existing ligand is of limited use to address these issues. However, if generally applicable this approach may be useful for other relevant target proteins or for the de novo design of inhibitors. For that reason, I see potential for publication in Nat. Commun. I do not have the expertise to evaluate the computational workflow and therefore focused on the introduction and experimental parts. Overall, the manuscript makes a preliminary impression with central experimental data and information missing. See the following list for details:

The authors write: "All of the tested compounds, including PNU-74654, UU-T01, UU-T02, MSAB, HI-B1, and 4FNPC had K_d values in the micromolar range, indicating that none of these primary hits are sufficiently potent." For many of those compounds, binding to beta-catenin could not be reproduced (see, McCoy J. Med. Chem. 2022, 65, 7246).

Author reply: We agree with the reviewer. We have added the reference and revised the text accordingly.

Considering that the manuscript reports peptide ligands, it would be more appropriate to cite alternative confirmed peptide-inspired ligands of beta-catenin (e.g. Schneider, Nat. Commun. 2018, 9, 4396; Wendt, Angew. Chem. Int. Ed. 2021, 60, 13937; Blosser, ACS Chem. Biol. 2021, 16, 1518).

Author reply: We fully agree with the reviewer. We have cited these references and discussed them in the revised manuscript.

The authors write: "As a transcription factor, the β -catenin also features a significant degree of structural disorder." This is not correct since i) beta-catenin is not a transcription factor but a transcriptional coactivator, and ii) structural disorder is not a prerequisite for a transcription factor (neither for a transcriptional coactivator). Also, the authors refer to a stapled peptide in the next sentence, which binds to the folded armadillo repeat domain of beta-catenin and not its flexible regions. This should be clarified.

Author reply: We thank the reviewer for pointing out this mistake. We have revised it in the revised manuscript with the following sentence:

"Rational design of small-molecule β -catenin inhibitors has proven to be challenging due to the large and flat surface of β -catenin lacking a well-defined binding pocket."

The authors refer to an example in which a CPP was fused to afore mentioned stapled peptide and write that “both binding affinity and specificity ... were compromised” (reference 5). Here, the study should be mentioned that reports a cell permeable CPP-modified peptide that shows high affinity and selectivity for beta-catenin (Ditrich, Cell Chem. Biol. 2017, 24, 958).

Author reply: We thank the reviewer for this comment. The reference has been added.

Readability of Figure 1 would improve if consistent colour code was used (e.g. same color for beta-catenin). Also, it is not clear why the CPP modification is shown and not just the original crystal structure.

Author reply: All figures have been redrawn and improved.

The SI should contain analytics of all synthesized peptides and plots of raw data that was used to calculate reported IC50 values.

Author reply: Analytical data for synthesized peptides including purity assessment by HPLC/UPLC and high-resolution MS are provided in SI. Raw data from FP-based competition assays are also included.

With respect to the in silico affinity maturation of peptide ligands the authors should describe the state of the art (here just one example: Rooklin, J. Am. Chem. Soc. 2017, 139, 15560).

Author reply: We have added the following new paragraph to the revised manuscript.

Numerous computational approaches have been developed to design target-specific peptide inhibitors [12-15]. These methods can be broadly divided into two classes: (a) structure-based and (b) sequence-based methods. The structure-based approaches start the design from a protein pocket or an existing peptide motif bound to the protein. For example, Rosetta FlexPepDock that combines extensive conformational search and a template-based strategy has been proven effective in modeling a wide array of peptide-protein complexes [16-18]. Rooklin et al. proposed a computational method to identify pockets near the peptide motif, and then design inhibitors that optimize pocket occupation [19]. However, starting from a structural template could bias or limit the sequence space search, leading to suboptimal solutions [20]. On the other hand, purely sequence-based methods have not been widely used in peptide design. Strikingly, the recently emerged AlphaFold that is transforming the field of protein structure prediction has shown a remarkable success in identifying high affinity peptide binders from a set of peptides [21, 22]. However, a major caveat of AlphaFold is that it remains computationally demanding to be combined with sequence search for peptide design. Additionally, the AlphaFold-based design is limited to natural amino acids and also incompatible with popular peptide affinity maturation methods, including crosslinking, cyclization or scaffolding.

The authors write: “...peptide 9 bound to β -catenin with an IC50 value of 145 ± 35 ” What is the unit of this value?

Author reply: The unit is nanomolar. We apologize for the error.

Readability of Table 1 would be improved if sequences were aligned and a monospaced font would be selected.

Author reply: These have been improved in the revised manuscript.

The authors should give a bit of background regarding one-bead-one compound libraries (e.g. Lam lab and many others).

Author reply: We have provided the additional information in the revised manuscript as requested. Under Section 4.7, a 1997 review by Lam and co-workers has been cited when describing the split-and-pool synthesis of OBOC peptide library (page 29, line 9).

Experimental peptide affinities have only been assessed using FP companion assays. The authors should perform an alternative assay (e.g. ITC) and also determine KD values in a direct FP with fluorescently labelled peptides.

Author reply: Since the parent peptide (peptide 9) is a well-established ligand of β -catenin, we felt that binding analysis by a single method is sufficient to establish the relative affinities of the N- or C-terminally extended peptides. FP-based competition assay is very robust and a better choice than the direct FP assay in this case, as the latter can be biased by the labeling dye.

The authors could test activity of their peptides in cell-based assays when using CPP-modified versions (as done by the study serving as the starting point).

Author reply: Since cellular activity is affected by many other factors, including metabolic stability, such data may or may not help the main message of this work, i.e., our AI approach can quickly discover peptide binders with improved binding affinity, irrespective of their other properties. Thus, we chose not to test activity of these peptides in cell-based assays in this study.

Point-to-point response to Reviewer #2:

The paper by Chen et al. reports a generative algorithm for the generation of optimized peptides for binding to B-catenin. They used a known binder of B-catenin and used their machine learning method to extend the N-terminal of the peptide. Followed by Rosetta FlexPepDock and MD, they picked some peptides to experimentally test and showed that some of their peptides achieve 2x enhanced binding. They also showed that a random extended library does not perform as well. Overall the reviewer thinks that there is not enough evidence that this method is in fact a powerful approach to creating better binders given the marginal improvement in binding with the need for so much computational work. Since the method only applied to one example of a helical binder that has been extended, the reviewer cannot see how it can be generalized to more complicated cases where such starting motif does not exist.

Author reply: We thank the reviewer for this critical comment. We have expanded our study to design C-terminally extended peptide inhibitors against β -catenin. 4 out of the 8 designed C-terminal extensions show improved IC₅₀ values, with the best one CAL-2 being 150-fold more

potent than the parent peptide. We believe these new results provide strong evidence that our computational approach can indeed design better target-specific peptide binders.

The reviewer also finds the flow of the paper a little confusing due to inconsistencies across. Examples:

- 1. Some tables use $\times 10^2$ nM (which is a strange choice) and some use μ M*
- 2. They mention their best AI-based peptide has 84 nM IC₅₀. But the table shows $8.4 \times 10^2 = 840$ nM.*
- 3. The parent peptide used for the comparison of synthetic library has different binding affinity than the one used for the computational one.*
- 4. They refer to structural info in figure 3C-E but it seems like they meant 1C-E*

Author reply: We apologize for these errors, and have fixed all of them in the revised manuscript.

The rationale behind why they did what they did and whether it truly holds promise is questionable to the reviewer. Here are some areas of question:

- 1. The best binder from AI method has an IC₅₀ of 84 nM. It seems like their control set (Table 1) already includes better binders and many of them are obtained via substitutions in C-terminal.*

Author reply: It is true that our control set already includes a few potent binders with their IC₅₀ values in the range of ~30 nM. However, these peptides contain substitutions of unnatural amino acids that cannot be predicted by our deep learning model due to insufficient training data. Nevertheless, if compared to the baseline parent peptide, the best of our designed N-terminally extended peptide showed a 2-fold affinity improvement. Additionally, we have expanded our model to the C-terminal extension in the revision, and designed 8 new C-terminally extended peptides. Four of them exhibit enhanced affinity for β -catenin, with the best one showing an IC₅₀ of 10 nM, which is approximately 150-fold better than the truncated parent peptide. We have revised the text with these new results.

- 2. The authors use a number of scores from Rosetta FlexPepDock to decide N-terminal extension works better than C-terminal extension. They also used these measures as selection criteria in other parts of their work. However, nowhere in the paper one can see a plot showing that these scores or their combination reflect experimental binding affinity (they have a plot for their MD method, 7A). Without that, the use of these metrics doesn't feel justified.*

Author reply: We have added a correlation plot (Figure 3a in the SI) between the Rosetta FlexPepDock scores and the experimental affinity data for a small set of peptides. However, we don't think this result is very meaningful as the Rosetta FlexPepDock scores or their combinations are not expected to correlate well with the experimental affinity data. Nevertheless, like molecular docking scoring function or molecular mechanics force field, even though individual energy scores are not accurate enough for pose or conformation prediction, the general trend should hold – the “correct” conformations or poses should generally occupy the low energy regions. Thus, the key idea here is to use Rosetta FlexPepDock to enrich “good” peptides. We are only interested in those peptides with high Rosetta FlexPepDock scores that will have a high probability of being good

binders. Eventually, 6 out of the 12 designed peptides (50%) showed improved affinity over the parent peptide. We believe these results support the use of these Rosetta FlexPepDock metrics as a first-layer filter to prioritize peptides for more rigorous calculation.

3. In figure 5, all the sampled AI models as well as the random model show better values than the control set except I_bsa. Given that there are some very good binders in the control set, the reviewer is curious why this is happening. It seems like perhaps the inclusion of the two outliers (peptides 12 and 13 with very high IC50) might have skewed the distribution. One would expect random extensions don't work better than controls overall unless any extension is better than no extension, which does not sound plausible.

Author reply: We thank the review for pointing out this important caveat of the Rosetta FlexPepDock scores. The Rosetta Flexpepdock scores generally favor longer or larger peptides due to the increased nonspecific interactions. Given that both AI sampled models and random models are all extended and a few residues longer than the control set, it is thus not very surprising that all the sampled models show better Rosetta values than the control set. As a result, it would not be very meaningful to compare the absolute Rosetta Flexpepdock scores for drastically different sequences. So in the revised manuscript, we have presented data of the relative Rosetta Flexpepdock scores – the Rosetta Flexpepdock score differences between the extended peptides and the parent peptide – only the relevant contribution from the extensions (Figures 3 and 6).

4. Comparing the AI model with just random sequences seems unfair to the reviewer given that there are faster methods with which one can generate extensions. For example, the authors could have generated extended peptides using Rosetta and used Rosetta Design to assign sequences. That would be a better comparison in terms of state-of-the-art compared to random. Especially given that their model is much slower (due to the requirement of MD step)

Author reply: We apologize for the confusion. We did not compare the AI model with random sequences, instead used Rosetta Design for a baseline comparison with our AI model. The comparison shows that VAE-MH performs significantly better than Rosetta Design with more favorable I_sc and I_bsa. This result is now plotted in Figure 3c.

5. The authors did a great job documenting their code. However, details of the AI model are not clear in the text. For example, it is unclear what the fixed size for the GRU unit is. If the unlabeled sequences are at 50 and only 15 is kept as the base, it means that 35 residues will be used for extension. However, 35 is really far from the true extension value used here, 2-5 AA.

Author reply: We apologize for the ambiguity. Our extensions have a varying length but are all in the range of 2-9 amino acid residues. For the N-terminal extension, 2-5 amino acid residues are added. For the C-terminal extensions, 2-7 and 4-9 residues are added to the parent peptide with the terminal residues IL and the truncated parent peptide without the terminal IL, respectively. We have clarified these descriptions in the revised manuscript.

6. The reviewer wonders why the authors did not pick a random subset of AI-generated sequences for experimental library synthesis to compare the performance. It also would be good to get either a visual or mathematical summary of the library screening.

Author reply: The AI-generated peptides are expected to have a low probability of binding to β -catenin as the AI model was not trained with sufficient β -catenin binding data. Thus, it is not very meaningful to test a random subset of AI-generated sequences, which would take a lot of time and resources. Nevertheless, to increase the statistical significance of our model, we performed additional predictions of the C-terminal extensions and synthesized 8 new peptides. Four of these 8 peptides showed improved affinity compared to the parent peptide with the best one being 150-fold more potent than the truncated parent peptide. These results validate the performance of our AI model. We have added Figure 5 to show the experimental library screening results.

7. How do the authors verify that their random UniProt sample does not include binders? Many proteins bind to other proteins for function.

Author reply: We agree with the reviewer that many proteins bind to other proteins for function. However, the probability of a random protein interacting with another random protein would be extremely small. For a rough estimation, let's consider the recently constructed Human Reference Interactome (HuRI) map that charts 52,569 interactions between 8,275 human proteins (Luck et. Al., Nature, 2020, 580, 402-408). This gives an upper limit of the interaction probability of ~0.15% given that indirect or transient interactions were not excluded in the study. Therefore, it is highly unlikely that our random UniProt sample would contain any β -catenin binders.

8. The differences in Rosetta FlexPepDock scores do not seem significantly meaningful to the reviewer based on their experience. It's also unclear if p-value of 0.1 is calculated given the big error that MM/GBSA has with regards to predicting true binding affinities and whether that shows any significant difference between AI VAE2, GA and SA.

Author reply: We agree with the reviewer that neither Rosetta FlexPepDock nor MM/GBSA is very accurate for binding affinity prediction. However, it has been demonstrated that Rosetta FlexPepDock can distinguish active peptides from inactive peptides, and MM/GBSA can rank the peptides well as evidenced by the good correlation between the MM/GBSA ranks and the IC_{50} ranks for a set of 14 experimentally tested peptides (Figure 3b in SI). Therefore, as stated in our reply to Comment #3, our goal is not to predict binding affinities for these peptides, but rather to enrich good peptides that will be prioritized for experimental testing through a combined Rosetta FlexPepDock and MM/GBSA two-layer filtering strategy. Given that 6 of the 12 AI-designed peptides (50%) showed improved affinities, we feel that our two-layer filtering strategy worked the way it was supposed to.

The figures can improve:

1. It's really hard to get any conclusion about figures 1C-E about structure (authors referred to those for justification of extension)

Author reply: The figure has been redrawn.

2. The reviewer is curious why the convention of using violin plots was not used in fig. 5. The lines need to be thicker for that figure.

Author reply: The figure has been redrawn and improved.

3. In figure 9, are the starting point the predicted docked peptides from FlexPepDock? Also the reviewer thinks “Structural Analysis” can be a little misleading of a title for 2.8 as it suggests that there are real structures. You only know these are models by reading the figure caption.

Author reply: We thank the reviewer for pointing out this confusion. The starting structure for the MD simulation is from the *FlexPepDock* refined structure. We have clarified these statements in the revised manuscript.

Reviewers' Comments:

Reviewer #1:

Remarks to the Author:

Overall, the revised manuscript by Chen et al. has improved both with respect to writing and figures. Most of the reviewer points have been addressed. However, two important aspects have been neglected:

1) Reviewer 2: "Since the method only applied to one example of a helical binder that has been extended, the reviewer cannot see how it can be generalized to more complicated cases where such starting motif does not exist."

The authors explain that extending the peptide further using their approach would be sufficient to show generality.

2) This reviewer: "The authors could test activity of their peptides in cell-based assays when using CPP-modified versions."

The authors argue that the study aims for increasing binding affinity, and that cellular activity is therefore not of interest.

This reviewer disagrees with the authors and thinks it is essential to address at least one of the two aspects with additional experiments or calculations:

1) If the authors see the main contribution of this study in the computational approach, this should be stronger emphasized in title and introduction. Also, the approach should then be applied for at least one more peptide sequence (on a different beta-catenin binding site or on a different protein).

2) It is important to note that the cellular delivery of beta-catenin targeting peptides is the major hurdle for their applicability. The authors decided to give a detailed introduction to the Wnt signaling pathway and also mention "targeting beta-catenin" in the title. In this context, this reviewer considers the validation of cellular activity of new inhibitors crucial.

We appreciate the reviewer's constructive comments and suggestions. Presented below are our point-to-point responses to the reviewer's comments.

Point-to-point response to Reviewer #1:

This reviewer disagrees with the authors and thinks it is essential to address at least one of the two aspects with additional experiments or calculations:

*1) If the authors see the main contribution of this study in the computational approach, this should be stronger emphasized in title and introduction. Also, the approach should then be applied for at least one more peptide sequence (on a different beta-catenin binding site or on a different protein).
2) It is important to note that the cellular delivery of beta-catenin targeting peptides is the major hurdle for their applicability. The authors decided to give a detailed introduction to the Wnt signaling pathway and also mention "targeting beta-catenin" in the title. In this context, this reviewer considers the validation of cellular activity of new inhibitors crucial.*

Author reply: We thank the reviewer for the critical comments and valuable recommendations. We have adopted the reviewer's suggestion 1 by making the following modifications:

1. We fully concur with the reviewer's assessment that the primary contribution of our study lies in the computational approach. To align with this point, we have modified the title and introduction of our manuscript to emphasize the computational aspect of our work.

2. To expand the scope of our computational approach, we have extended its application to design N-terminally extended peptide inhibitors against nuclear factor- κ B (NF- κ B) essential modulator (NEMO). Compared to β -catenin, NEMO presents a more challenging system for peptide inhibitor design due to the dynamic and weak interaction between NEMO and the 11-residue parent peptide, the NEMO binding domain (NBD) of IKK β . NBD showed very weak binding with an IC₅₀ \gg 100 μ M. Despite these challenges, our research has yielded promising results, with two out of the four designed N-terminal extensions displaying substantially enhanced binding compared to NBD, with IC₅₀ values of approximately 50 μ M and 75 μ M, respectively.

We believe these new findings together with the prior results (now encompassing three sets of designed peptides targeting three distinct binding sites across two proteins) constitute compelling evidence of the effectiveness and versatility of our computational approach in the design of highly potent target-specific peptide binders.

Once again, we greatly appreciate the reviewer's insightful comment and valuable recommendation, which has significantly helped improve our manuscript.

Reviewers' Comments:

Reviewer #1:

Remarks to the Author:

With the adjustment of the title and additional computational as well as experimental data, the authors have addressed the reviewer comment appropriately. Recommendation: publish without alterations.